Proceedings of the 7th Symposium on Advances in Approximate Bayesian Inference, 2025 1–42

# Massively Parallel Expectation Maximization For Approximate Posteriors

**Thomas Heap**                                              THOMAS.HEAP@BRISTOL.AC.UK
**Sam Bowyer**                                                SAM.BOWYER@BRISTOL.AC.UK
**Laurence Aitchison**                                  LAURENCE.AITCHISON@GMAIL.COM
*University of Bristol, UK*

## Abstract

Bayesian inference for hierarchical models can be very challenging. MCMC methods have difficulty scaling to large models with many observations and latent variables. While variational inference (VI) and reweighted wake-sleep (RWS) can be more scalable, they are gradient-based methods and so often require many iterations to converge. Our key insight was that modern massively parallel importance weighting methods (Bowyer et al., 2024) give fast and accurate posterior moment estimates, and we can use these moment estimates to rapidly learn an approximate posterior. Specifically, we propose using expectation maximization to fit the approximate posterior, which we call QEM. The expectation step involves computing the posterior moments using high-quality massively parallel estimates from Bowyer et al. (2024). The maximization step involves fitting the approximate posterior using these moments, which can be done straightforwardly for simple approximate posteriors such as Gaussian, Gamma, Beta, Dirichlet, Binomial, Multinomial, Categorical, etc. (or combinations thereof). We show that QEM is faster than state-of-the-art, massively parallel variants of RWS and VI, and is invariant to reparameterizations of the model that dramatically slow down gradient based methods.

## 1. Introduction

Hierarchical models are powerful statistical tools for understanding real-world, structured data. For instance, a hierarchical model could be used to analyze student test scores, where students are nested within classrooms and classrooms are nested within schools. However, using Bayesian inference in these hierarchical models to estimate the underlying parameters or latent variables from the data can be computationally challenging. MCMC methods such as HMC (Duane et al., 1987; Neal, 2012) have difficulty scaling to large hierarchical models, so there is a large literature applying variational inference (VI; Jordan et al., 1999; Blei et al., 2017; Aitchison, 2019; Agrawal and Domke, 2021; Geffner and Domke, 2022) and reweighted wake-sleep (RWS; Bornschein and Bengio, 2015; Le et al., 2020; Heap et al., 2023) as an alternative.

However, with VI and RWS, in the case of non-conjugate models the approximate posterior is often learned using gradient descent on an objective function (Kingma and Welling, 2014; Rezende et al., 2014; Bornschein and Bengio, 2015; Le et al., 2020; Wingate and Weber, 2013; Titsias and Lázaro-Gredilla, 2014). Gradient descent can require many iterations to converge, requires careful hyperparameter tuning, and is sensitive to details of how the model is parameterized (see Appendix E).

In this paper, we set out to develop a Bayesian inference method for hierarchical models which is faster than both VI and RWS, while at the same time mitigating those methods' issues, such as inefficient gradient-based training and overconfident approximate posteriors. To develop this method, we began with massively parallel importance weighting (MPIW) (Bowyer et al., 2024). To motivate MPIW, consider a model with data $x$, latent variables $z'$, true posterior, $P(z'|x)$ and approximate posterior, $Q(z')$. (Note that we use $z'$ rather than $z$ as we will use $z$ later to represent the collection of samples of the latents used in importance weighting.) Chatterjee and Diaconis (2018) showed that for importance weighting, the number of samples from the proposal, $Q(z')$, required to get a good estimate of the true posterior, $P(z'|x)$, scales as $\exp(D_{KL}(P(z'|x)\|Q(z')))$. This exponential scaling of the number of samples looks problematic. Indeed, Bowyer et al. (2024) note that the KL-divergence scales linearly with the number of latent variables, $n$, implying that the number of samples required scales as $\exp(n)$, which is intractable in all but the smallest models. Massively parallel methods resolve this issue by drawing $K$ samples for each of the $n$ latent variables, and explicitly reweighting all possible $K^n$ combinations. MPIW is able to be efficient because they exploit conditional independencies in the generative model and proposal/approximate posterior. These conditional independencies are usually understood by writing both distributions as a "Bayes net". Thus, MPIW in effect has the required exponential number of importance samples, and can give good posterior moment estimates even in large models.

However, MPIW is ultimately just an importance sampling method, i.e. it allows you to sample from a single, fixed proposal then reweight to approximate the true posterior. To work well, all importance sampling methods require a proposal distribution that is reasonably close to the true posterior. This motivates considering iterative schemes that learn better proposals. Indeed, pre-existing methods such as VI (Jordan et al., 1999; Wainwright et al., 2008; Kingma and Welling, 2014; Rezende et al., 2014; Blei et al., 2017; Nguyen et al., 2018; Zhang et al., 2018; Kingma et al., 2019; Gayoso et al., 2021) and reweighted wake-sleep (Bornschein and Bengio, 2015; Le et al., 2020) do just this—using gradient ascent to update the proposal (which is usually called an approximate posterior in the VI/RWS context).

To develop an alternative, we noticed that we could use MPIW posterior moment estimates to learn the proposal without needing gradient descent. We refer to this new method as QEM, as it resembles the expectation maximization algorithm (Dempster et al., 1977; Neal and Hinton, 1998). The key difference is that traditional EM optimizes the parameters of the *prior*, whereas QEM optimizes the parameters of the *approximate posterior* (the approximate posterior is denoted Q in VI and RWS, hence the QEM acronym). Specifically, in the expectation step (E-step) we use MPIW to compute posterior moments, and in the maximization step (M-step) we use these moments to update the approximate posteriors. Note that the M-step is straightforward for simple approximate posteriors such as Gaussian, Gamma, Beta, Dirichlet, Binomial, Multinomial, Categorical, etc. Critically, standard EM differs from QEM because in standard EM, there is no approximate posterior. Instead, standard EM learns parameters of the model $\theta$ which may appear in the prior $P_\theta(z')$ or in the likelihood, $P_\theta(x|z')$.

We show that QEM converges more rapidly than VI or RWS, is reparameterization invariant, gives moment estimates that improve over VI, and are comparable to those in RWS.

## 2. Related work

Of course, key related work arises in the small but growing literature on massively parallel methods, including massively parallel VI (Aitchison, 2019) and massively parallel RWS (Heap et al., 2023), along with massively parallel importance weighting (Bowyer et al., 2024). This work all uses the key massively parallel idea—drawing $K$ samples of each of the $n$ latent variables and reasoning over all $K^n$ combinations—in different contexts.

Massively parallel VI (Aitchison, 2019) is an extension of the importance weighted autoencoder (IWAE; Burda et al., 2016). IWAE learns a good approximate posterior by doing gradient ascent on a multi-sample variant of the ELBO (Eq. 14c) that gives tighter variational bounds. However, IWAE only uses a small number of importance samples from the approximate posterior to compute the ELBO. Unfortunately, given the results from Chatterjee and Diaconis (2018) discussed in the introduction, we expect the number of importance samples required to accurately approximate the true posterior to scale exponentially in number of latent variables. This suggests that in larger models, IWAE will have far too few samples to approximate the true posterior. Massively parallel VI resolves this issue by drawing $K$ samples for each of the $n$ latent variables, and computing the ELBO by averaging over all $K^n$ combinations (Eq. 7c). Aitchison (2019) showed that massively parallel VI converges faster than standard VI, and gives considerably tighter bounds on the marginal likelihood.

Likewise, massively parallel RWS (Heap et al., 2023) is a massively parallel extension of RWS (Bornschein and Bengio, 2015; Le et al., 2020). To understand RWS, note that the ideal way to learn the approximate posterior would be maximum likelihood on samples from the true posterior. Of course, we do not have samples from the true posterior, so what else can we do? One approach is to approximate the true posterior by sampling from the approximate posterior and reweighting. Of course, this procedure requires a good importance weighted estimate of the true posterior. However, traditional RWS uses only a small number of approximate posterior samples, which, as previously discussed, may not suffice in settings with a large number of latent variables. Massively parallel RWS (Heap et al., 2023) resolves this issue by again drawing $K$ samples of each latent variable and considering all $K^n$ combinations.

Importantly, both massively parallel VI and RWS improve the approximate posterior by doing gradient ascent, which can be slow and requires tuning hyperparameters such as the learning rate. In contrast, QEM introduced here uses massively parallel moment estimates to directly update the approximate posterior, without using gradients.

Massively parallel importance weighting (MPIW) was introduced in Bowyer et al. (2024). We use MPIW to compute moment estimates for QEM. Importantly, however, QEM and MPIW are doing something quite different. MPIW is just an importance weighting method, so it allows you to sample from a proposal/approximate posterior, then reweight to approximate the true posterior. Importantly, the proposal/approximate posterior in importance sampling is fixed. In contrast, QEM is a method to learn a better approximate posterior.

As QEM learns a better approximate posterior, it can be understood as an adaptive importance sampling/weighting method (Ortiz and Kaelbling, 2000; Liu et al., 2001; Pennanen and Koivu, 2006; Rubinstein and Kroese, 2004; Cornuet et al., 2012; Marin et al., 2014; Bugallo et al., 2015, 2017; El-Laham et al., 2019; Elvira et al., 2019; van Osta et al., 2021). We

believe that these methods could be enhanced by incorporating massively parallel importance weighted (MPIW) moment estimates (Bowyer et al., 2024), but we leave this complex integration to future work.

QEM is invariant to reparameterization of the approximate posterior, which resembles properties you might see in natural gradient variational inference (Amari, 1998; Hoffman et al., 2013; Hensman et al., 2013; Khan and Nielsen, 2018). However, natural gradient inference is restricted to performing variational inference in the single-sample setting, so can be expected to give loose bounds on the ELBO and poor moment estimates (Turner and Sahani, 2011; Burda et al., 2016; Aitchison, 2019; Geffner and Domke, 2022).

## 3. Background

### 3.1. Bayesian inference.

We have a probabilistic model with latents $z'$ and data $x$, with a prior, $P(z')$ and a likelihood, $P(x|z')$. Our goal is to compute properties of the posterior distribution over latent variables, $z'$, conditioned on data, $x$. This posterior is given by Bayes theorem,

$$P\left(z'|x\right) = \frac{P\left(x|z'\right)P\left(z'\right)}{P\left(x\right)}, \tag{1}$$

where $P(x)$ is known as the marginal likelihood or model evidence,

$$P\left(x\right) = \int dz' P\left(x|z'\right)P\left(z'\right). \tag{2}$$

In QEM, we need to compute posterior moments,

$$m^{\text{true}} = \mathbb{E}_{P(z'|x)}\left[m(z')\right]. \tag{3}$$

where $m(z')$ is any moment of interest. These moments are usually intractable and must be approximated.

### 3.2. Massively parallel importance weighting.

Traditional importance sampling methods fail to obtain a good estimate of the posterior for hierarchical models as the required number of samples scales exponentially with the number of latent variables. MPIW resolves this by drawing $K$ samples for each of the $n$ latent variables and explicitly reweighting all $K^n$ combinations. For each latent variable $i$, we draw $K$ samples:

$$z_i = (z_i^1, \ldots, z_i^K) \in \mathcal{Z}_i^K, \tag{4}$$

and the collection of all samples of all latents is given by

$$z = (z_1, \ldots, z_n) \in \mathcal{Z}^K. \tag{5}$$

There are then two types of approximate posterior distributions:

---

**Algorithm 1** Expectation maximization for approximate posteriors (QEM)

---

$m_0$ = Initial values. {Initialise approximate posterior mean parameters.}
**for** $t \in 1, 2, \ldots, T$ **do**
  $Q_t$ = set_exp_family_params($m_{t-1}$) {Set exponential family parameters (M-step)}
  $z \sim Q_t$
  $m_t^{\text{one iter}}$ = MPIW($z, Q_t$) {Compute moments using MPIW (E-step)}
  $m_t = \lambda m_t^{\text{one iter}} + (1 - \lambda) m_{t-1}$ {Update mean parameters via EMA}
**end for**

---

- $Q(z)$: The standard approximate posterior distribution over a single complete set of latent variables

- $Q_{\text{MP}}(z)$: The massively parallel approximate posterior that handles all $K^n$ combinations of samples, where $n$ is the number of latent variables

While $Q(z)$ works with just one set of latents, $Q_{\text{MP}}(z)$ is structured to handle the full combinatorial space created by having $K$ samples for each latent variable. Specifically, $Q_{\text{MP}}(z)$ factorizes according to the graphical model structure:

$$Q_{\text{MP}}(z) = \prod_{i=1}^{n} Q_{\text{MP}}(z_i | z_j \text{ for } j \in \text{qa}(i)) \tag{6}$$

where $\text{qa}(i)$ represents the parents of the $i$th latent variable in the graphical model for the approximate posterior. This factorization allows $Q_{\text{MP}}(z)$ to efficiently handle the massive number of combinations while respecting the conditional independence structure of the model. More details on this can be found in Appendix A.

Given indices $\mathbf{k} = (k_1, \ldots, k_n) \in \mathcal{K}^n$, we compute the MPIW moment estimate:

$$m_{\text{MP}}(z) = \frac{1}{K^n} \sum_{\mathbf{k} \in \mathcal{K}^n} \frac{r_{\mathbf{k}}(z)}{\mathcal{P}_{\text{MP}}(z)} m(z^{\mathbf{k}}) \tag{7a}$$

$$r_{\mathbf{k}}(z) = \frac{P(x, z^{\mathbf{k}})}{\prod_i Q_{\text{MP}}\left(z_i^{k_i} \middle| z_j \text{ for } j \in \text{qa}(i)\right)} \tag{7b}$$

$$\mathcal{P}_{\text{MP}}(z) = \frac{1}{K^n} \sum_{\mathbf{k} \in \mathcal{K}^n} r_{\mathbf{k}}(z). \tag{7c}$$

Heap et al. (2023) showed that Eq. (7c) is an unbiased estimator of the marginal likelihood. Computing these estimates efficiently requires exploiting conditional independencies in the graphical model (Bowyer et al., 2024).

While this resembles standard importance weighting, MPIW sums over all $K^n$ combinations of samples for all latent variables. Bowyer et al. (2024) showed these moment estimates can be computed efficiently using the source-term trick—differentiating through a modified version of the ELBO—by exploiting conditional independencies in the graphical model. Heap et al. (2023) showed that $\mathcal{P}_{\text{MP}}(z)$ provides an unbiased estimate of the marginal likelihood.

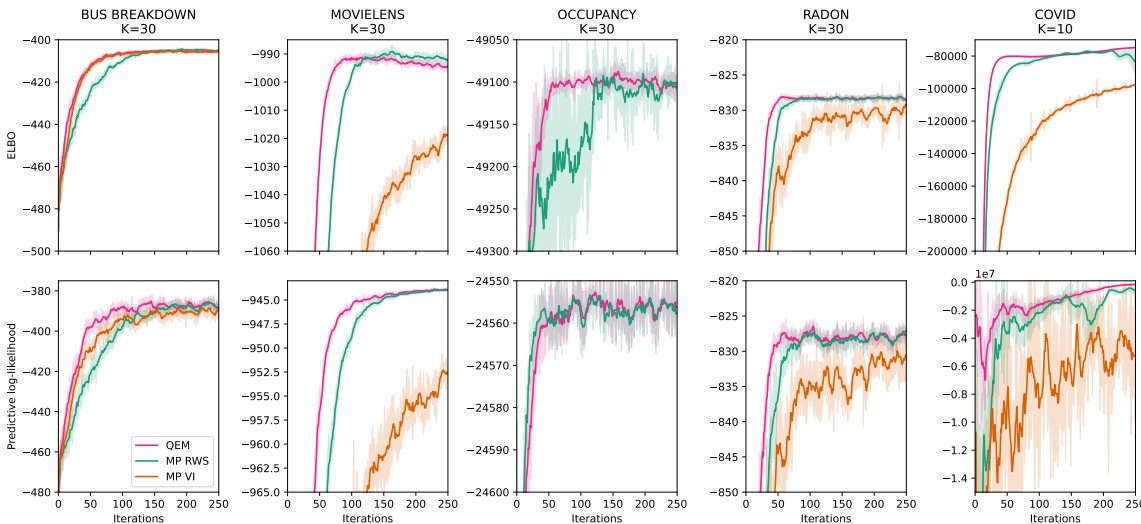

Figure 1: Comparing the ELBO (top row) and predictive-log-likelihood (bottom row) of QEM (pink), RWS (green) and VI (orange) on several models, with iteration number on the x-axis. We report error bars on each line of one standard error over five repeated runs with the same data but using different random seeds. Note we did not run VI on the occupancy model as it has discrete latent variables.

## 4. Methods

Here, we introduce expectation maximization for approximate posteriors (QEM). The overall approach is to alternate between E and M steps:

- E-step: use MPIW to estimate true posterior moments.

- M-step: compute the natural/conventional parameters of the approximate posterior by setting the moments of the approximate posterior equal to the (average of) moments computed in the E-step.

The algorithm operates by setting parameters of $Q(z)$ based on current moment estimates, then using $Q_{MP}(z)$ to compute more accurate estimates by exploiting the full combinatorial space of samples. This separation allows us to maintain a simple parameterization while leveraging the power of massively parallel importance weighting.

For an efficient M-step, we assume that the approximate posterior is an exponential family distributions such as Gaussian, Beta, and Dirichlet in which the distributions natural or conventional parameters can be recovered directly from the moments. For instance, a Gaussian component's mean and variance are determined by its first and second moments, while a Beta's parameters are recovered from expectations of log transformations. See Appendix B for an in-depth discussion of the relationship between the QEM M-step and RWS.

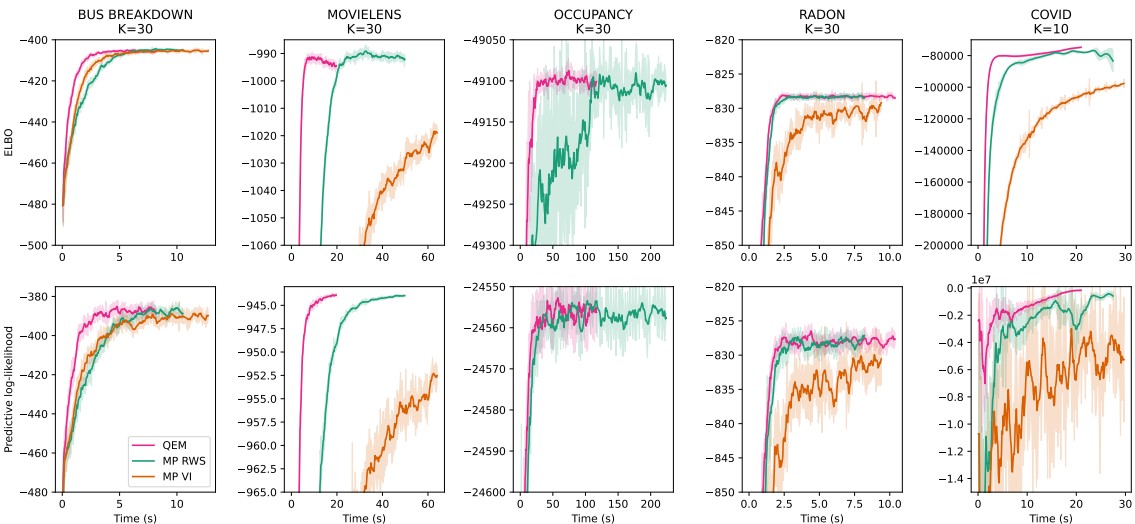

Figure 2: As in Figure 1, but with time, rather than iterations on the x-axis. Again, note we cannot run VI on the occupancy model as it has discrete latent variables.

In practice, just using the previous moment estimate can cause pathologies (e.g. if the second moment estimates on one step happen to be very close to the first moment estimates due to only one of the samples having non-negligible weight). We can reduce these pathologies by using an exponential moving average (EMA) for the moments. Specifically, if $m_{t-1}$ is the previous EMA moment estimate, and $m_t^{\text{one iter}}$ is the new MPIW moment estimate, then we would form a new EMA moment estimate using

$$m_t = (1 - \lambda)m_{t-1} + \lambda m_t^{\text{one iter}}. \tag{8}$$

Algorithm 1 presents the complete QEM procedure. In each iteration, the algorithm uses the current mean parameters $m_{t-1}$ to set the parameters of exponential family distributions that form the approximate posterior $Q_t$ (line 3). This step exploits the fact that exponential family distributions are completely determined by their mean parameters—for instance, a Gaussian's mean and variance can be derived from its first and second moments. The algorithm then draws samples from $Q_t$ (line 4) and uses MPIW to compute accurate estimates of the true posterior moments (line 5). Finally, these moment estimates are combined with previous estimates using an exponential moving average (line 6) to reduce the impact of any single noisy estimate.

Importantly, QEM differs from standard EM (Dempster et al., 1977) because in standard EM, there is no approximate posterior. Instead, standard EM learns parameters of the model $\theta$ which may appear in the prior $P_\theta(z')$ or in the likelihood, $P_\theta(x|z')$. The QEM approach provides several advantages: it avoids gradient-based optimization, is invariant to reparameterization (Section 5.2), and converges to the correct moments as the number of importance samples increases (Appendix C). Of course, approaches such as MC-EM (Wei and Tanner, 1990; Levine and Casella, 2001) use a MCMC approximation to the posterior.

However, MCMC can be very slow, and may parallelise poorly. In contrast, we use fast and highly parallelisable massively parallel methods.

While this approach might seem natural, there are actually a number of choices here. For instance, at each step we could use MPIW to estimate the moments (in an exponential family, the mean parameters), then we could immediately compute the corresponding conventional parameters (for a Gaussian, the mean and variance) then we could do an EMA for these conventional parameters. Likewise, we could also compute the EMA of the estimated exponential family natural parameters. We choose the mean parameters, because you only get the same fixed points as massively parallel reweighted wake-sleep if we do the EMA for the mean parameters (Appendix B).

Indeed, we can prove that if you have an unbiased estimator of the parameter of interest, many iterations and a sufficiently large $\lambda$, then the EMA gives the right answer,

**Theorem 1** *Consider an exponential moving average moment estimator of the form Eq. (8), where $m_t^{one\ iter}$ is an unbiased estimator with finite variance and where*

$$\lambda(t) = 1 - t^{-p}, \tag{9}$$

*with $0 < p < 1$. In the limit as $t \to \infty$, $m_t$ is unbiased, and zero variance,*

$$\lim_{t \to 0} \text{Var}\,[m_t] = 0. \tag{10}$$

See D for a proof. Note that MPIW, as presented in Equation 7a-7c, gives such an unbiased estimator $m_t^{\text{one iter}}$ in the limit as $K \to \infty$, since it is a self-normalizing importance weighting strategy (meaning the same sample $z$ is used to calculate $\mathcal{P}_{\text{MP}}(z)$ as well as $r_{\mathbf{k}}(z)$ and $m(z^{\mathbf{k}})$). We can achieve an unbiased $m_t^{\text{one iter}}$ with finite $K$ if we simply use a second, distinct sample $z_{\text{new}} \sim Q_{\text{MP}}$ to calculate $\mathcal{P}_{\text{MP}}(z_{\text{new}})$ instead (with $r_{\mathbf{k}}(z)$ and $m(z^{\mathbf{k}})$ unchanged).

## 5. Results

We consider the following five datasets: Bus Breakdown (Heap et al., 2023), MovieLens100K (Harper and Konstan, 2015), Occupancy (Bowyer et al., 2024), Radon (Gelman and Hill, 2006) and Covid (Leech et al., 2022). Bus Breakdown consists of around 150,000 New York city bus delays stratified by year and borough, we use a subset consisting of 2 years, 3 boroughs in each year and 300 delays in each borough. Movielens100K consists of 100,000 movie ratings from 943 users of 1682 movies. The Radon dataset consists of radon readings taken at houses in the United States, of which we take a subset of 4 states and 300 readings in each state. The Bird Occupancy dataset comes from the North American Breeding Bird Survey, which reports the number of sightings of over 700 species of birds from 1966 to 2021 along thousands of roadside routes in the USA and Canada. The Covid dataset is adapted from (Leech et al., 2022) and consists of 137 daily Covid-19 infection numbers from 92 countries.

For each dataset we define a hierarchical generative model $P\left(x, z^{\mathbf{k}}\right)$ and an approximate posterior/proposal distribution $Q_{\text{MP}}\left(z^{\mathbf{k}}\right)$ over the same latents $z$. See Appendix F for full details of the priors and approximate posteriors. For the approximate posteriors, we usually use a Gaussian independent of the other latent variables.

We use strong baselines: massively parallel variational inference (MP VI; Aitchison, 2019) and massively parallel reweighted wake-sleep (MP RWS; Heap et al., 2023). For MP VI and MP RWS, we need an optimizer. We use the Adam optimizer (Kingma and Ba, 2014) with a learning rate given by selecting the value from 0.3, 0.1, 0.03, 0.01, 0.003, 0.001 which led to the greatest ELBO after 125 iterations (which is also how we chose $\lambda$ for the EMA). All other Adam hyperparameters are left at their PyTorch defaults. For VI the objective is simply the massively parallel ELBO, (Eq. 7c), whilst for RWS this is a slightly modified massively parallel ELBO (as discussed in Heap et al., 2023) which performs a maximum-likelihood update on the parameters of $Q_{MP}$ using posterior samples obtained via proposal samples reweighted by the importance weights $r_{\mathbf{k}}(z)$ from Eq. (7b).

To measure the quality of inference achieved we consider three metrics. First, we use the importance weighted ELBO. This can be seen as a single-sample ELBO with an improved approximate posterior (Cremer et al., 2017; Bachman and Precup, 2015), and since the single-sample ELBO directly measures the KL divergence between the true posterior and approximate posterior,

$$\text{ELBO} = \log P(x) - D_{KL}(Q(z') \,||\, P(z|x)), \tag{11}$$

the importance weighted ELBO is commonly taken as a proxy for the quality of the importance weighted posterior estimate (Geffner and Domke, 2022; Agrawal and Domke, 2021; Domke and Sheldon, 2019). When interpreting the ELBO, it is important to remember that VI directly maximizes the ELBO, and while QEM and MP RWS do not directly maximize the ELBO; nonetheless, it remains a useful diagnostic for all methods. Secondly, we use predictive log-likelihood: we draw latent samples conditioned on "training" data from each dataset and use these to predict a separate "test" set. Details about the train and test splits for each dataset can be found in their respective sections in the appendix. Thus, in expectation a higher predictive log-likelihood indicates importance samples which are closer to the true posterior. Thirdly, we simply measure the mean-squared error between posterior first moment estimates and "true" posterior first moments (which we find using long HMC runs).

We use $K = 30$ in all experiments except for the Covid dataset in which we use $K = 10$ due to memory constraints which apply equally to QEM and the massively parallel baselines (MP VI and MP RWS). In each experiment we run 250 iterations of the specified inference algorithm to optimize the approximate posterior.

### 5.1. Model comparison

We begin by comparing QEM against its closest competitors: MP RWS and MP VI. Fig. 1 shows performance as measured by the ELBO (top) and predictive log-likelihood (bottom) versus iterations on five datasets. Fig. 2 mirrors Fig. 1, but has time on the x-axis. Note that occupancy has a discrete latent variable and so therefore doesn't have a differentiable likelihood, precluding the possibility of using MP VI or HMC. Figure 1 shows that QEM outperforms MP VI in terms of both ELBO and predictive log-likelihood in every model, and learns faster than MP RWS in all settings except Radon and Occupancy when considering the predictive log-likelihood. Also notice that the standard errors (shown in the error bars of Figures 1 and 2) of QEM's ELBO and predictive log-likelihood results are typically much

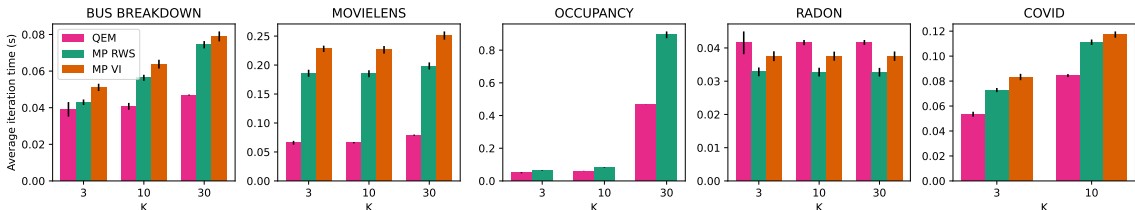

Figure 3: Comparing the time-per-iteration between QEM (pink), RWS (green) and VI (orange) on several models with varying values of K. Black error bars represent one standard deviation over all iterations and experiment repeats.

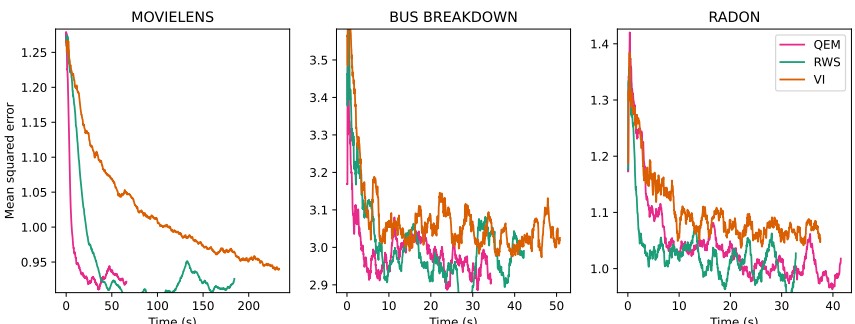

Figure 4: Mean squared error between first moment estimates for each method and HMC first moment estimates plotted against time. Occupancy is not plotted because it has discrete latent variables, preventing the use of HMC, and Covid is not plotted because we were not able to scale HMC to this larger model.

lower than that of the other methods. The plots against time are even more stark: notice that since we do not need to calculate gradients for the whole joint distribution, QEM runs significantly faster than MP VI and MP RWS.

This can be seen more clearly in Figure 3: the mean time taken to complete a single iteration of QEM is almost always less than MP VI and MP RWS for all values of $K$, with the only exception occurring in the Radon model, where all methods are extremely fast but QEM is very slightly slower than the others.

We also show in Figure 4 that the quality of posterior moments/approximate posterior obtained by QEM is superior to that of MP VI and at least as good as MP RWS, as measured by the mean squared distance to the true posterior means, obtained by HMC via the BlackJAX NUTS sampler (Cabezas et al., 2024).

### 5.2. QEM is invariant under reparameterization

One final advantage of QEM is that it is invariant to reparameterizations which leave the distribution over the data intact, e.g. $z' = \alpha \tilde{z}$ for some constant factor $\alpha$ on a latent variable $\tilde{z}$. To test this, we rerun each of our experiments but modify the models by scaling down the size of a single latent variable by a factor of $\alpha \in \{1/100, 1/1000, 1/100{,}00\}$, before scaling it back up by $1/\alpha$ for use later in the model—thus leaving the rest of the overall distribution unchanged. (Note that these reparameterizations are not related to the reparameterization trick.)

Figure 5 shows that whilst QEM retains identical performance under these changes (barring occasional small numerical differences from floating point operations at the different scales), the gradient-based updates for MP VI and MP RWS are not invariant to such reparameterizations, and therefore experience slower, noisier optimization and numerical instability (Fig. 5 shows VI failing before reaching 250 iterations on the reparameterized Radon and Covid models). See Appendix E for a proof and more details on the experimental setting.

## 6. Limitations

QEM faces the same main limitations as other massively parallel methods: complexity of implementation and memory constraints. As mentioned in Section 3, computing the massively parallel posterior moment estimates is, in general, far from trivial. However, using the strategy developed in Bowyer et al. (2024), this is made possible (and efficient) for a very large class of probabilistic models. Similarly, the task of reasoning over an effectively exponential number of samples presents immediate concerns for memory consumption. In order to deal with this we follow the strategy adopted in Aitchison (2019) to reduce the memory complexity from $O(K^n)$ to $O(K^{\max_i |\mathrm{qa}(i)|})$ where $|\mathrm{qa}(i)|$ is the number of parent random variables for the $i$th child random variable in a massively parallel proposal $Q_{\mathrm{MP}}$. Further implementation details can also be used such as grouping random variables together in the proposal and splitting necessary computations into manageable chunks. Such details as they pertain to the experiments performed in this paper can be found in Appendix F.

## 7. Conclusion

We present a new approach for performing Bayesian inference on general probabilistic graphical models, based on performing expectation maximization on an approximate posterior using massively parallel posterior moment estimates, described in Section 4. We have shown in Section 5 that this new method, QEM, not only outperforms other standard techniques such as massively parallel VI and RWS in terms of higher ELBOs, predictive log-likelihoods and more accurate posterior moments, but also takes less time to do so thanks to its gradient-free approach. Furthermore, we present theoretical justifications and empirical results for QEM's superior robustness to model reparameterization in comparison to gradient-based methods such as VI and RWS in Section 5.2 and Appendix E.

We see QEM as a significant development in displaying the power of massively parallel probabilistic inference. As future work, we anticipate more inference methods making use of

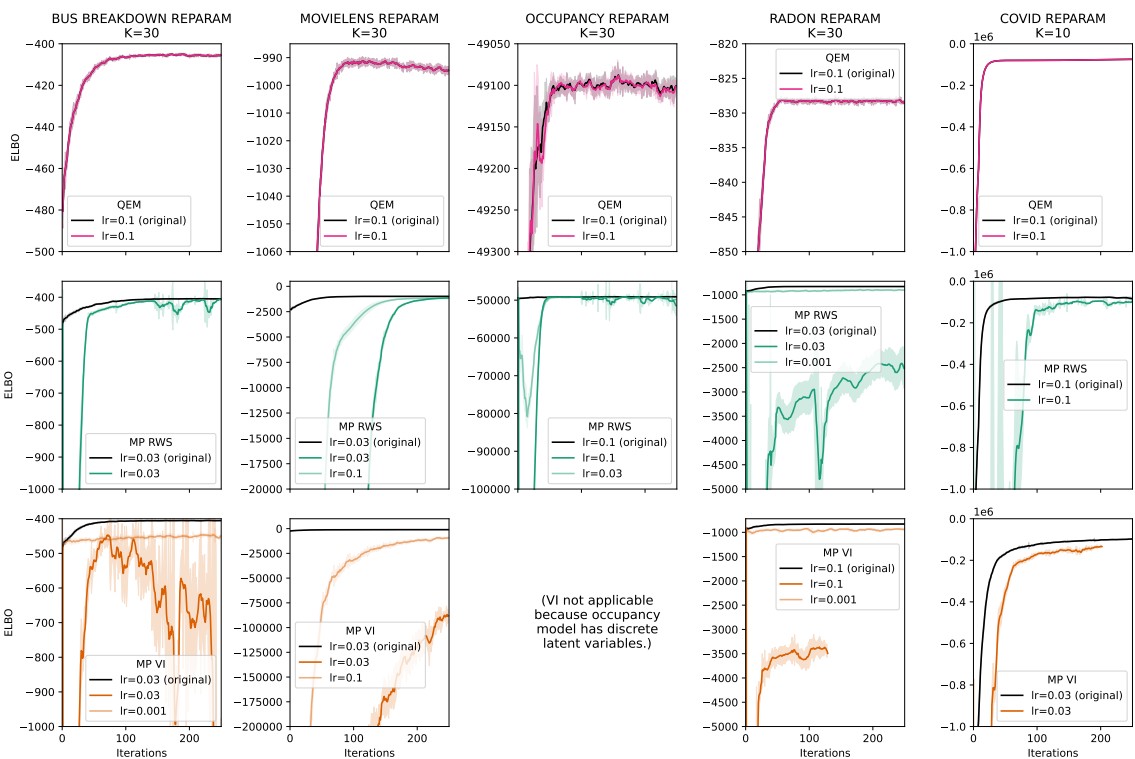

Figure 5: Comparing the ELBOs achieved by each method on reparameterized models (coloured lines) versus the original parameterization (black lines), with error bars representing standard errors over five runs with the same data but using different random seeds. In many cases, the reparameterization led to a different learning rate being optimal for MP VI and MP RWS. Where this occurred, we have plotted the learning rate that was optimal for the original parameterization in a solid colour and the learning rate that was optimal for the reparameterization in a fainter colour. Note: our implementation of VI relies on the reparameterization trick and thus doesn't work for models with discrete latents.

the massively parallel framework, with the aim to collect these into a complete probabilistic programming language.

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

## Appendix A. Additional Background Details

### A.1. "Global" importance weighting.

Importance weighting is an approach to approximating posterior moments. The standard approach to importance weighting, called "Global" in Geffner and Domke (2022) is to sample $K$ times from the approximate posterior. We use $z^k \in \mathcal{Z}$, for a single sample from the full joint latent space, and use

$$z = (z^1, z^2, \ldots, z^K) \in \mathcal{Z}^K. \tag{12}$$

for the collection of all $K$ samples. We sample $z$ by drawing $K$ times from the approximate posterior,

$$Q(z) = \prod_{k \in \mathcal{K}} Q\left(z^k\right). \tag{13}$$

Here, $\mathcal{K}$ is the set of indices for the samples, $\mathcal{K} = \{1, \ldots, K\}$. The self-normalized global importance weighted estimate is,

$$m_{\text{global}}(z) = \tfrac{1}{K} \sum_{k \in \mathcal{K}} \frac{r_k(z)}{\mathcal{P}_{\text{global}}(z)} m(z^k) \tag{14a}$$

$$r_k(z) = \frac{P\left(x, z^k\right)}{Q\left(z^k\right)} \tag{14b}$$

$$\mathcal{P}_{\text{global}}(z) = \tfrac{1}{K} \sum_{k \in \mathcal{K}} r_k(z). \tag{14c}$$

Here, $r_k(z)$ is the joint probability of $z^k$ and $x$ under the generative model divided by the probability of $z^k$ under the approximate posterior, and $\mathcal{P}_{\text{global}}(z)$ is an unbiased estimator of the marginal likelihood,

$$E_{Q(z)}\left[\mathcal{P}_{\text{global}}(z)\right] = E_{Q\left(z^k\right)}\left[\frac{P\left(x, z^k\right)}{Q\left(z^k\right)}\right] = \int dz^k \; P\left(x, z^k\right) = P(x).$$

The first equality arises by taking Eq. (14c), and noticing that each $r_k(z)$ term is IID.

### A.2. Massively parallel importance weighting.

In standard "global" importance sampling above we reweighted $K$ samples from the full joint latent space. But in massively parallel methods, we exploit the fact that we have multiple latent variables. Write $z_i^k \in \mathcal{Z}_i$ for the $k$th sample of the $i$th latent variable. Thus all $K$ samples of the $i$th latent variable are written

$$z_i = (z_i^1, \ldots, z_i^K) \in \mathcal{Z}_i^K \tag{15}$$

and the collection of all samples of all latents is given by

$$z = (z_1, \ldots, z_n) \in \mathcal{Z}^K. \tag{16}$$

Given indices $\mathbf{k} = (k_1, \ldots, k_n) \in \mathcal{K}^n$, we can therefore represent a single sample of the latents as:

$$z^{\mathbf{k}} = (z_1^{k_1}, \ldots, z_n^{k_n}) \in \mathcal{Z}. \tag{17}$$

Thus, $z$ is all $K$ samples for all $n$ latents. $z_i$ is all $K$ samples for the $i$th latent. $z^k$ is the $k$th sample of all $n$ latents. $z_i^k$ is the $k$th sample for the $i$th latent.

The massively parallel setting requires a slightly modified proposal distribution. In particular, we define $Q_{\mathrm{MP}}(z)$ with graphical model structure

$$Q_{\mathrm{MP}}(z) = \prod_{i=1}^{n} Q_{\mathrm{MP}}(z_i|z_j \text{ for } j \in \mathrm{qa}(i)), \tag{18}$$

such that $\mathrm{qa}(i)$ is the set of indices of parents of $z_i$ under the graphical model. The conditional distribution $Q_{\mathrm{MP}}(z_i|z_j \text{ for } j \in \mathrm{qa}(i))$ over all $K$ copies of the $i$th and $j$th latent variables, i.e. $z_i$ and $z_j$, is defined via a single-sample proposal (much like in the "global" setting) where single copies of the $i$th and $j$th latents, $z_i'$ and $z_j'$, are distributed as $Q\left(z_i' \middle| z_j' \text{ for } j \in \mathrm{qa}(i)\right)$. This required a scheme for choosing which of the $K$ copies of a parent latent gets used in the sampling of each $K$ copy of the child latent. As discussed in Heap et al. (2023), many such schemes are possible, such as a uniform mixture over the $K$ parent samples or a permutation (chosen uniformly at random) of the $K$ parent samples. (In all experiments performed for this paper the permutation strategy was used.)

## Appendix B. The QEM M-step and massively parallel reweighted wake-sleep

Perhaps the optimal approach for fitting an approximate posterior would be maximum likelihood, under samples drawn from the true posterior. That would give an objective of the form

$$\mathcal{L}_{\mathrm{post}} = \mathrm{E}_{\mathrm{P}(z'|x)}\left[\log Q\left(z'\right)\right], \tag{19}$$

which gives updates of the form,

$$\Delta\eta = \epsilon \mathrm{E}_{\mathrm{P}(z'|x)}\left[\nabla_\eta \log Q\left(z'\right)\right], \tag{20}$$

where $\eta$ are the parameters of the approximate posterior, and $\epsilon$ is a small learning rate.

However, we cannot directly use these updates, as they require samples from the true posterior. Instead, we can use an importance-weighted approximation to the updates,

$$\Delta\eta = \epsilon \, \frac{1}{K^n} \sum_{\mathbf{k} \in \mathcal{K}^n} \frac{r_{\mathbf{k}}(z)}{\mathcal{P}_{\mathrm{MP}}(z)} \nabla_\eta \log Q\left(z^{\mathbf{k}}|x\right). \tag{21}$$

where $z$ is sampled from $Q(z)$. These updates are used by massively-parallel reweighted wake-sleep (Heap et al., 2023).

For exponential family approximate posteriors,

$$\log Q\left(z'\right) = \eta \cdot T(z') - A(\eta) \tag{22}$$

where $\eta$ is the natural parameters, $T(z')$ is the sufficient statistics, and $A(\eta)$ is the log-normalizing constant. Now, differentiate with respect to $\eta$,

$$\nabla_\eta \log Q\left(z'\right) = T(z') - \nabla_\eta A(\eta) \tag{23}$$

where,

$$\nabla_\eta A(\eta) = \nabla_\eta \log \int dz' \ \exp\left(\eta \cdot T(z')\right) \tag{24}$$

$$= \frac{\int dz' \ T(z') \exp\left(\eta \cdot T(z')\right)}{\int dz' \ \exp\left(\eta \cdot T(z')\right)} \tag{25}$$

$$= \mathrm{E}_{Q(z')}\left[T(z')\right] \tag{26}$$

$$= \mu \tag{27}$$

are the mean-parameters of $Q\left(z'\right)$. Thus,

$$\nabla_\eta \log Q\left(z'\right) = T(z') - \mu. \tag{28}$$

And we can write the updates as,

$$\Delta\eta(z) = \epsilon\left(m^{\text{one iter}}(z) - \mu\right). \tag{29}$$

where,

$$m^{\text{one iter}}(z) = \frac{1}{K^n} \sum_{\mathbf{k} \in \mathcal{K}^n} \frac{r_{\mathbf{k}}(z)}{\mathcal{P}_{\text{MP}}(z)} T(z') \tag{30}$$

is a moment estimate based on a single sample, $z$ of $K$ samples for all latent variables. Here, the first term is the expected sufficient statistics of the massively parallel importance weighted approximation of the true posterior, while the second term is the expected sufficient statistics of the approximate posterior itself. Thus, for sufficiently small learning rates, $\epsilon$, standard massively parallel reweighted wake sleep is at a fixed point when the approximate posterior sufficient statistics are equal to the massively parallel importance weighted sufficient statistics,

$$\mu = \mathrm{E}_{Q(z)}\left[m^{\text{one iter}}(z)\right] \tag{31}$$

This provides the inspiration for the QEM M-step updates, which directly set the expected sufficient statistics of the approximate posterior to an the importance-weighted estimate of the true posterior moments (which thus has the same fixed points). Specifically, the QEM M-step updates are,

$$\Delta\mu = \lambda\left(m^{\text{one iter}}(z) - \mu\right), \tag{32}$$

which obviously have the same fixed-point.

Note that these QEM M-step updates are equivalent to an EMA (Eq. 8). Importantly, we would not get the same fixed points if we took an EMA over another quantity. For instance, if we took $\eta^{\text{one iter}}$,

$$\Delta\eta = \lambda\left(\eta^{\text{one iter}}(z) - \eta\right), \tag{33}$$

would imply a fixed point of,

$$\eta = \mathrm{E}_{\mathrm{Q}(z)} \left[ \eta^{\text{one iter}}(z) \right] \tag{34}$$

This is not the same as the fixed point in Eq. (31) as the mapping from natural, $\mu$, to mean, $\eta$, parameters is nonlinear.

## Appendix C.  Convergence of QEM in the limit as K goes to infinity

As $K \to \infty$, (massively parallel) importance sampling gives the right moments under mild conditions (Robert et al., 1999), such as the support of $\mathrm{Q}(z')$ covering the support of $\mathrm{P}(z'|x)$. In that setting, QEM converges in a single step.

## Appendix D.  Proof of Theorem 1

We can write the EMA in Eq. (8) as,

$$m_t = (1 - \lambda) \sum_{\tau=1}^{t} \lambda^{t-\tau} m_\tau^{\text{one iter}}. \tag{35}$$

As $m_\tau^{\text{one iter}}$ is unbiased, we have $\mathrm{E} \left[ m_\tau^{\text{one iter}} \right] = m^{\text{true}}$,

$$\mathrm{E}\left[m_t\right] = (1 - \lambda) \sum_{\tau=1}^{t} \lambda^{t-\tau} m^{\text{true}}. \tag{36}$$

Using the standard expression for geometric series,

$$\mathrm{E}\left[m_t\right] = (1 - \lambda)\frac{1 - \lambda^t}{1 - \lambda} m^{\text{true}} = \left(1 - \lambda^t\right) m^{\text{true}}. \tag{37}$$

Consider $\lambda$ increasing as $t$ increases,

$$\lambda(t)^t = (1 - t^{-p})^t. \tag{38}$$

To evaluate this limit, we use composition of limits[1], with

$$\lim_{t\to\infty} \lambda(t)^t = \lim_{t\to\infty} \exp\left(t \log(1 - t^{-p})\right) \tag{39}$$

We can compute the limit inside the exponential using the L'Hopital's rule.

$$\log \lambda(t)^t = t \log(1 - t^{-p}) \tag{40}$$

$$\lim_{t\to\infty} t \log(1 - t^{-p}) = \lim_{t\to\infty} \frac{\log(1 - t^{-p})}{t^{-1}} \tag{41}$$

$$= \lim_{t\to\infty} \frac{\frac{pt^{-p-1}}{1-t^p}}{-t^{-2}} \tag{42}$$

---

1. e.g. https://mathcenter.oxford.emory.edu/site/math111/limitsOfCompositions/

Using $p < 0 < 1$,

$$= \lim_{t \to \infty} -pt^{-p+1} = \infty. \tag{43}$$

Thus,

$$\lim_{t \to \infty} \lambda(t)^t = \exp\left(-\log \lambda(t)^t\right) = 0 \tag{44}$$

and,

$$\lim_{t \to \infty} \mathrm{E}\left[m_t\right] = m^{\mathrm{true}}. \tag{45}$$

$$\tag{46}$$

To compute the variance of $m_t$, we need

$$\mathrm{Var}\left[m_t\right] = \mathrm{E}\left[(m_t - \mathrm{E}\left[m_t\right])^2\right] \tag{47}$$

$$= (1 - \lambda)^2 \mathrm{E}\left[\left(\sum_{\tau=1}^{t} \lambda^{t-\tau}(m_\tau^{\mathrm{one\ iter}} - m^{\mathrm{true}})\right)\left(\sum_{\tau'=1}^{t} \lambda^{t-\tau'}(m_{\tau'}^{\mathrm{one\ iter}} - m^{\mathrm{true}})\right)\right]. \tag{48}$$

Pushing the expectation inside the sum,

$$\mathrm{Var}\left[m_t\right] = (1 - \lambda)^2 \sum_{\tau=1}^{t}\sum_{\tau'=1}^{t} \lambda^{2t-\tau-\tau'} \mathrm{E}\left[(m_\tau^{\mathrm{one\ iter}} - m^{\mathrm{true}})(m_{\tau'}^{\mathrm{one\ iter}} - m^{\mathrm{true}})\right]. \tag{49}$$

All the terms for $\tau \neq \tau'$ are zero. Taking only the terms for which $\tau = \tau'$ and using $\mathrm{E}\left[(m_\tau^{\mathrm{one\ iter}} - m^{\mathrm{true}})^2\right] = \mathrm{Var}\left[m_\tau^{\mathrm{one\ iter}}\right] = v$,

$$\mathrm{Var}\left[m_t\right] = (1 - \lambda)^2 \sum_{\tau=1}^{t} \lambda^{2(t-\tau)}v. \tag{50}$$

As this is again a geometric series,

$$\mathrm{Var}\left[m_t\right] = (1 - \lambda)^2 \sum_{\tau=1}^{t} \left(\lambda^2\right)^{t-\tau} v, \tag{51}$$

we have,

$$\mathrm{Var}\left[m_t\right] = (1 - \lambda)^2 \frac{1 - \lambda^{2t}}{1 - \lambda^2}v. \tag{52}$$

We begin by noting (by analogy with the derivation for the mean),

$$\lim_{t \to \infty} \lambda(t)^{2t} = \lim_{t \to \infty} (1 - t^{-p})^{2t} = 0, \tag{53}$$

so $\lim_{t\to\infty} 1 - \lambda(t)^{2t} = 1$, and

$$\lim_{t\to\infty} \text{Var}\left[m_t\right] = \frac{(1-\lambda)^2}{1-\lambda^2} v. \tag{54}$$

And as,

$$\lim_{t\to\infty} \lambda(t) = \lim_{t\to\infty} \left(1 - t^{-p}\right) = 1, \tag{55}$$

we have,

$$\lim_{t\to\infty} \text{Var}\left[m_t\right] = \lim_{\lambda\to 1} \frac{(1-\lambda)^2}{1-\lambda^2} v. \tag{56}$$

Applying L'Hopital's rule,

$$\lim_{t\to\infty} \text{Var}\left[m_t\right] = \lim_{\lambda\to 1} \frac{-2(1-\lambda)}{2\lambda} v = 0. \tag{57}$$

Thus, the variance converges to zero.

However, it does not establish that reducing the KL-divergence will definitely reduce the variance and reduce the required number of samples. For that result, we look to Chatterjee and Diaconis (2018).

## Appendix E. QEM is invariant to reparameterization

Consider "reparameterizing" a model by rewriting the original latent variable, $z'$

$$\tilde{z}' = \tfrac{1}{\alpha} z', \tag{58}$$

as an alternative latent variable, $\tilde{z}'$, which is related to the original latent variable by a multiplicative scaling.

The original model is given by,

$$\text{P}\left(z'\right) = \pi(z') \tag{59a}$$
$$\text{P}\left(x|z'\right) = f(x, z') \tag{59b}$$

where $\pi(z')$ and $f(x, z')$ are functions giving the probability for the prior and likelihood. We get exactly the same distribution over the data if we reparameterize by setting $z' = \alpha\tilde{z}'$,

$$\text{P}\left(\tilde{z}'\right) = \alpha\pi(\alpha\tilde{z}') \tag{60a}$$
$$\text{P}\left(x|\tilde{z}'\right) = f(x, \alpha\tilde{z}') \tag{60b}$$

Now, consider a family of exponential family approximate posterior densities, $\text{Q}\left(z'; m\right)$ where $m$ is mean the parameters (e.g. for a Gaussian, $m$ is the raw first and second moment). This family must be closed under multiplication, in the sense that for $z'$ and $\tilde{z}'$ related by Eq. (58), there must exist $m$ and $\tilde{m}$ such that,

$$z' \sim \text{Q}\left(z'; m\right), \tag{61a}$$
$$\tilde{z}' \sim \text{Q}\left(\tilde{z}'; \tilde{m}\right). \tag{61b}$$

The parameters of the distributions are related by a function, $f$,

$$m = f(\tilde{m}, \alpha). \tag{62}$$

For instance, if we have a Gaussian, then

$$m = \begin{pmatrix} \mu \\ \mu^2 + \sigma^2 \end{pmatrix} \tag{63}$$

$$\tilde{m} = \begin{pmatrix} \tilde{\mu} \\ \tilde{\mu}^2 + \tilde{\sigma}^2 \end{pmatrix} \tag{64}$$

$$f\left( \begin{pmatrix} \tilde{\mu} \\ \tilde{\mu}^2 + \tilde{\sigma}^2 \end{pmatrix}; \alpha \right) = \begin{pmatrix} \alpha \tilde{\mu} \\ \alpha^2 \left( \tilde{\mu}^2 + \tilde{\sigma}^2 \right) \end{pmatrix} \tag{65}$$

Note that while most standard distributions that are members of scale or location-scale families that are closed under multiplication, not all are, with perhaps the easiest example being the Poisson.

Finally, we need reparameterization-invariant sampling, in the sense that, sampling proceeds by transforming an IID random variable, $\xi$,

$$z' = g(\xi; m) \sim Q\left( z'; m \right), \tag{66}$$
$$\tilde{z}' = g(\xi; \tilde{m}) \sim Q\left( \tilde{z}'; \tilde{m} \right). \tag{67}$$

such that,

$$g(\xi; m) = \alpha g(\xi; \tilde{m}) \tag{68}$$

if $m = f(\tilde{m}, \alpha)$. This property holds for common sampling algorithms, e.g. when using inverse transform sampling.

**Theorem 2** *QEM is invariant to reparameterization, if three properties hold:*

- *The model is reparameterized in the sense of Eq. (59) and Eq. (60).*

- *Sampling is reparameterized in the sense of Eq. (68).*

- *The approximate posterior family is closed under multiplication, in the sense of Eq. (61) and Eq. (62).*

*In the sense that if the initial parameters of the approximate posterior are reparameterized,*

$$m_0 = f(\tilde{m}_0, \alpha), \tag{69}$$

*then the approximate posterior parameters, $m_t$ and $\tilde{m}_t$ obtained by QEM at future timesteps remain reparameterized at all future timesteps,*

$$m_t = f(\tilde{m}_t, \alpha). \tag{70}$$

**Proof** Here, we do a proof by induction, showing that if the parameters are reparameterized $(m_t = f(\tilde{m}_t, \alpha))$ at timestep $t$, then they are also reparameterized at timestep $t+1$ (i.e. after a QEM update; $m_{t+1} = f(\tilde{m}_{t+1}, \alpha)$). The base case, $m_0 = f(\tilde{m}_0, \alpha)$, holds by assumption.

Now, if we start with a reparameterized approximate posterior, $m_t = f(\tilde{m}_t, \alpha)$, and sampling is reparameterized, then all $K$ samples of $n$ latent variables in QEM, are related by multiplication,

$$z_i'^k = \alpha z_i''^k. \tag{71}$$

Under the reparameterized model, the importance weights are exactly the same under the reparameterized model, as the importance weights depend only on $r_{\mathbf{k}}(z)$, the ratio of the generative to the approximate posterior probabilities,

$$r_{\mathbf{k}}(z) = \frac{P\left(x, z^{\mathbf{k}}\right)}{Q\left(z^{\mathbf{k}}\right)} = \frac{P\left(x, \tilde{z}^{\mathbf{k}}\right)}{Q\left(\tilde{z}^{\mathbf{k}}\right)} = r_{\mathbf{k}}(\tilde{z}) \tag{72}$$

because,

$$P\left(x, \tilde{z}^{\mathbf{k}}\right) = \alpha P\left(x, z^{\mathbf{k}}\right) \tag{73}$$

$$Q\left(x, \tilde{z}^{\mathbf{k}}\right) = \alpha Q\left(z^{\mathbf{k}}\right). \tag{74}$$

As the importance weights are the same and the samples are reparameterized, the moment estimates are reparameterized in the sense that,

$$m_{t+1}^{\text{one iter}} = f(\tilde{m}_{t+1}^{\text{one iter}}, \alpha). \tag{75}$$

Thus, if we just update the approximate posterior moments using the previous sample, then the inductive argument holds.

While it might seem that the extension to the EMA case is straightforward, it actually turns out to be surprisingly involved due to the nonlinearity of $f(\tilde{m}_{t+1}^{\text{one iter}}, \alpha)$. To prove that reparameterization holds after the EMA, we need to note that the EMA can be understood as computing the moments on a very large weighted sample. Specifically, substituting Eq. (30) into Eq. (8),

$$m_t = \frac{1-\lambda}{K^n} \sum_{\tau=1}^{t} \sum_{\mathbf{k} \in \mathcal{K}^n} \lambda^{t-\tau} \frac{r_{\mathbf{k}}(z)}{\mathcal{P}_{\text{MP}}(z)} T(z^{\mathbf{k}}). \tag{76}$$

This can be understood as a weight average over all samples at all timesteps. Again, the importance weights at all timesteps are the same, and the samples are reparameterized, so the moments are also reparameterized. ∎

To demonstrate this invariance empirically we learn approximate posteriors using QEM, MP RWS and MP VI for a modified version of each aforementioned model where some latent variables have been reduced by a factor of $10^{-1}, 10^{-2}, 10^{-3}$ or $10^{-4}$ (before being scaled back up for use later in the graphical model). The full reparameterized model specifications can be found in Appendix F.

Figure 5 shows that QEM retains identical performance (barring some small numerical differences in the occupancy model results) on these reparameterized models as on the original parameterizations. MP VI and MP RWS, on the other hand, struggle to reach the same ELBOs and predictive log-likelihoods as before, experiencing much noisier and slower optimization.

## Appendix F. Experimental datasets and models

### F.1. Experiment details

To obtain the results discussed in section 5, we use graphical models for each dataset, defining both a generative distribution $P\left(x, z^{\mathbf{k}}\right)$ and an approximate posterior/proposal distribution $Q_{\mathrm{MP}}\left(z^{\mathbf{k}}\right)$ where $x$ is data and $z^{\mathbf{k}}$ represents a single sample from the joint latent space. We then run 250 optimization steps of QEM, MP VI and MP RWS, at each iteration obtaining ELBOs, predictive log-likelihoods (averaged over 100 predictive samples) and posterior moment estimates using the massively parallel approach devised in Bowyer et al. (2024). This was repeated five times with different random seeds, with means and standard errors reported over these five runs. Similarly we repeated HMC runs five times using the BlackJAX NUTS implementation (Cabezas et al., 2024), using default hyperparameters. To adapt the parameters of NUTS we used 100, 300 and 1000 warm-up iterations, and drew 1000 samples for MovieLens and 3000 for both Bus Breakdown and Radon.

Table 1: Memory consumption and time comparison for QEM, MP RWS and MP VI on each of the models (with maximum $K$ values used per model).

| Model | Method | Memory Used (GB) | Time for 250 Iterations |
|---|---|---|---|
| Bus Breakdown $K = 30$ | QEM | 0.05 | 11.7s |
| | MP RWS | 0.06 | 18.6s |
| | MP VI | 0.06 | 19.7s |
| Movielens $K = 30$ | QEM | 1.10 | 19.8s |
| | MP RWS | 1.11 | 49.6s |
| | MP VI | 1.11 | 62.7s |
| Occupancy $K = 30$ | QEM | 8.80 | 116.9s |
| | MP RWS | 8.80 | 223.4s |
| Radon $K = 30$ | QEM | 0.01 | 13.4s |
| | MP RWS | 0.01 | 13.3s |
| | MP VI | 0.01 | 15.5s |
| Covid $K = 10$ | QEM | 2.76 | 13.4s |
| | MP RWS | 2.77 | 18.2s |
| | MP VI | 2.77 | 20.8s |

Learning rates for QEM, MP VI and MP RWS were chosen from $\{0.3, 0.1, 0.03, 0.01, 0.003, 0.001\}$ based on which led to the highest ELBO after 125 iterations.

For the MovieLens QEM, MP VI and MP RWS experiments, we split the dataset along the "user" dimension every 20 users during the calculation of approximate posterior updates in order to reduce the memory requirement at any one time during execution.

All experiments were performed on a 24GB NVIDIA RTX 3090, with memory consumption reported in Table 1 alongside the time taken to perform one run of 250 approximate posterior updates.

Note that more preliminary experiments were performed. In total, the compute time for all experiments used in generating the results of paper was approximately 250 GPU hours.

### F.2. Bus delay dataset

This experiment involves modelling New York school bus journeys delays,[2] using data supplied by the City of New York (DOE, 2023). Given the year, $y$, borough, $b$, bus company, $c$ and journey type, $j$, the aim is to predict whether a given delay is longer than 30 minutes. In particular, the data comprises 57 bus companies supplying 6 journey types (e.g. pre-K/elementary school route, general education AM/PM route etc.) over the years $2015 - 2022$ (inclusive) in the five New York boroughs (Brooklyn, Manhattan, The Bronx, Queens, Staten Island) and other surrounding areas (Nassau County, New Jersey, Connecticut, Rockland County, Westchester). We subsample the dataset at random to select $Y = 2$ years and $B = 3$ boroughs, constructing a training set of $I = 150$ delayed buses for each borough-year combination and an equally-sized test set of a further 150 delayed buses per combination on which to calculate predictive log-likelihood. As such, we can uniquely indentify each delay by the year, $y$, the borough, $b$, and the index, $i$, giving delay$_{ybi}$ The delays are recorded as an indicator: 1 if the delay is greater than 30 minutes, 0 otherwise.

To model the impact of each feature (year, borough, bus company and journey type) on the delaying of a bus, we use a hierarchical latent variable model in which the presence of a delay is modelled as a Bernoulli random variable. The expected presence of a delay is then described by a latent variable, logits$_{ybi}$, with one logit for each delayed bus. The logits are modelled as a sum of three terms which are themselves latent variables to be inferred: one for the borough and year jointly, one for the bus company and one for the journey type.

The first term considers the year and borough jointly: YearBoroughWeight$_{yb}$. This has a hierarchical prior: we begin by sampling a global mean and variance, GlobalMean and GlobalVariance; then for each year, we use GlobalMean and GlobalVariance to sample a mean for each year, YearMean$_y$. Additionally, we sample a variance for each year, YearVariance$_y$. Finally, we sample YearBoroughWeight$_{yb}$ from a Gaussian distribution with a year-dependent mean, YearMean$_y$, and variance $\exp(\text{BoroughVariance}_b)$.

The two other terms that contribute towards the logits are the weights for the bus company and journey type, which are modelled similarly. We have one latent weight for each bus company, CompanyWeight$_c$, with $c \in \{1, \ldots, 57\}$, and for each journey type, JourneyTypeWeight$_j$, with $j \in \{1, \ldots, 6\}$. Given the bus company, $b_{ybi}$, and journey type, $j_{ybi}$, for each delayed bus journey (remember that a particular delayed bus journey is uniquely

---

2. Dataset: http://data.cityofnewyork.us/Transportation/Bus-Breakdown-and-Delays/ez4e-fazm
   Terms of use: http://opendata.cityofnewyork.us/overview/#termsofuse

identified by the year, $y$, borough, $b$ and index $i$), we use a table to pick out the right company weight and journey type weight, $\text{CompanyWeight}_{c_{ybi}}$ and $\text{JourneyTypeWeight}_{j_{ybi}}$ which then contribute towards the sum giving us $\text{logits}_{ybi}$. The final generative model is given by

$$
\begin{aligned}
\mathrm{P}\left(\text{GlobalVariance}\right) &= \mathcal{N}(\text{GlobalVariance}; 0, 1) \\
\mathrm{P}\left(\text{GlobalMean}\right) &= \mathcal{N}(\text{GlobalMean}; 0, 1) \\
\mathrm{P}\left(\text{YearMean}_y | \text{GlobalMean}, \text{GlobalVariance}\right) &= \mathcal{N}(\text{YearMean}_y; \text{GlobalMean} \\
&\qquad \exp(\text{GlobalVariance})), \\
\mathrm{P}\left(\text{YearVariance}_b\right) &= \mathcal{N}(\text{YearVariance}_b; 0, 1) \\
\mathrm{P}\left(\text{YearBoroughWeight}_{yb} | \text{YearMean}_y, \text{YearVariance}_b\right) &= \mathcal{N}(\text{YearBoroughWeight}_{yb}; \\
&\qquad \text{YearMean}_y, \\
&\qquad \exp(\text{YearVariance}_b)) \\
\mathrm{P}\left(\text{CompanyWeight}_c\right) &= \mathcal{N}(\text{CompanyWeight}_c; 0, 1), \\
\mathrm{P}\left(\text{JourneyTypeWeight}_j\right) &= \mathcal{N}(\text{JourneyTypeWeight}_j; 0, 1) \\
\text{logits}_{ybi} &= \text{YearBoroughWeight}_{yb} \\
&\quad + \text{CompanyWeight}_{c_{ybi}} \\
&\quad + \text{JourneyTypeWeight}_{j_{ybi}} \\
\mathrm{P}\left(\text{delay}_{ybi} | \text{logits}_{ybi}\right) &= \text{Bernoulli}(\text{delay}_{ybi}; \text{logits}_{ybi}),
\end{aligned}
\tag{77}
$$

where $y \in \{1, \ldots, Y\}, b \in \{1, \ldots, B\}, c \in \{1, \ldots, C\}$ and $j \in \{1, \ldots, J\}$. The corresponding graphical model is given in Figure 6. We initialise the factorised approximate posterior $Q$ very similarly:

$$
\begin{aligned}
\mathrm{Q}\left(\text{GlobalVariance}\right) &= \mathcal{N}(\text{GlobalVariance}; 0, 1) \\
\mathrm{Q}\left(\text{GlobalMean}\right) &= \mathcal{N}(\text{GlobalMean}; 0, 1) \\
\mathrm{Q}\left(\text{YearMean}_y\right) &= \mathcal{N}(\text{YearMean}_y; 0, 1) \\
\mathrm{Q}\left(\text{YearVariance}_b\right) &= \mathcal{N}(\text{YearVariance}_b; 0, 1) \\
\mathrm{Q}\left(\text{YearBoroughWeight}_{yb}\right) &= \mathcal{N}(\text{YearBoroughWeight}_{yb}; 0, 1) \\
\mathrm{Q}\left(\text{CompanyWeight}_c\right) &= \mathcal{N}(\text{CompanyWeight}_c; 0, 1) \\
\mathrm{Q}\left(\text{JourneyTypeWeight}_j\right) &= \mathcal{N}(\text{JourneyTypeWeight}_j; 0, 1)
\end{aligned}
\tag{78}
$$

### F.2.1. Reparameterized bus delay model

To provide a reparameterized bus delay dataset we simply rescale the mean and standard deviation of the $\text{YearBoroughWeight}_{yb}$ latent variable, then apply the inverse scaling after sampling:

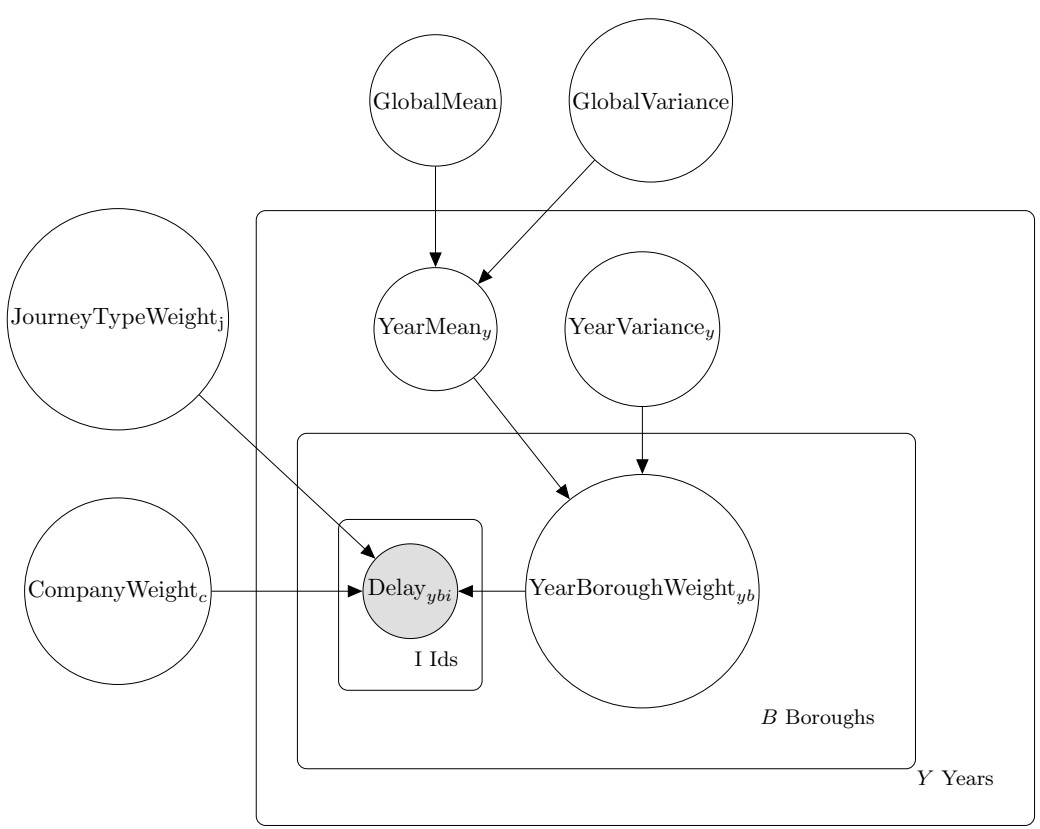

Figure 6: Graphical model for the NYC Bus Breakdown dataset

$$P\left(\text{GlobalVariance}\right) = \mathcal{N}(\text{GlobalVariance}; 0, 1)$$
$$P\left(\text{GlobalMean}\right) = \mathcal{N}(\text{GlobalMean}; 0, 1)$$
$$P\left(\text{YearMean}_y | \text{GlobalMean}, \text{GlobalVariance}\right) = \mathcal{N}(\text{YearMean}_y; \text{GlobalMean},$$
$$\exp(\text{GlobalVariance}))$$
$$P\left(\text{YearVariance}_b\right) = \mathcal{N}(\text{YearVariance}_b; 0, 1)$$
$$P\left(\text{YearBoroughWeight}_{yb} | \text{YearMean}_y, \text{YearVariance}_b\right) = \mathcal{N}(\text{YearBoroughWeight}_{yb};$$
$$\text{YearMean}_y * \frac{1}{1000},$$
$$\exp(\text{YearVariance}_b) * \frac{1}{1000})$$
$$P\left(\text{CompanyWeight}_c\right) = \mathcal{N}(\text{CompanyWeight}_c; 0, 1)$$
$$P\left(\text{JourneyTypeWeight}_j\right) = \mathcal{N}(\text{JourneyTypeWeight}_j; 0, 1)$$
$$\text{logits}_{ybi} = \text{YearBoroughWeight}_{yb} * 1000$$
$$+ \text{CompanyWeight}_{c_{ybi}}$$
$$+ \text{JourneyTypeWeight}_{j_{ybi}}$$
$$P\left(\text{delay}_{ybi} | \text{logits}_{ybi}\right) = \text{Bernoulli}(\text{delay}_{ybi}; \text{logits}_{ybi}), \tag{79}$$

and the approximate posterior distribution $Q$ is initialised as so:

$$
\begin{aligned}
\text{Q}\left(\text{GlobalVariance}\right) &= \mathcal{N}(\text{GlobalVariance}; 0, 1) \\
\text{Q}\left(\text{GlobalMean}\right) &= \mathcal{N}(\text{GlobalMean}; 0, 1) \\
\text{Q}\left(\text{YearMean}_y\right) &= \mathcal{N}(\text{YearMean}_y; 0, 1) \\
\text{Q}\left(\text{YearVariance}_b\right) &= \mathcal{N}(\text{YearVariance}_b; 0, 1) \\
\text{Q}\left(\text{YearBoroughWeight}_{yb}\right) &= \mathcal{N}(\text{YearBoroughWeight}_{yb}; 0, \frac{1}{1000}) \\
\text{Q}\left(\text{CompanyWeight}_c\right) &= \mathcal{N}(\text{CompanyWeight}_c; 0, 1) \\
\text{Q}\left(\text{JourneyTypeWeight}_j\right) &= \mathcal{N}(\text{JourneyTypeWeight}_j; 0, 1)
\end{aligned}
\tag{80}
$$

### F.3. Movielens dataset

The MovieLens100K[3] (Harper and Konstan, 2015) dataset contains 100k ratings from 0 to 5 of $N{=}1682$ films from among $M{=}943$ users. Following Geffner and Domke (2022), we binarise the ratings into likes and dislikes by mapping user-ratings of $\{0, 1, 2, 3\}$ to 0 (dislikes) and user-ratings of $\{4, 5\}$ to 1 (likes), and assume binarised ratings of 0 for films which users have not previously rated.

The probabilistic graphical model of the full generative distribution is given in Figure 7. We specify films using the index $n$ and specify users via the index $m$. Each film has a known feature vector, $\mathbf{x}_n \in \{0, 1\}^{18}$, indicating which of 18 genre tags (Action, Adventure, Animation, Children's, etc.) the film matches (each film may have multiple genre tags). Similarly, we model each user with a latent weight-vector, $\mathbf{z}_m \in \mathbb{R}^{18}$, describing whether or not they like any given feature. We model the probability of a user $m$ liking a film $n$ as $\sigma(\mathbf{z}_m^\mathsf{T}\mathbf{x}_n))$, where $\sigma(\cdot)$ is the sigmoid function. Thus the generative distribution may be written as

$$
\begin{aligned}
\text{P}\left(\boldsymbol{\mu}\right) &= \mathcal{N}(\boldsymbol{\mu}; \mathbf{0}_{18}, \mathbf{I}), \\
\text{P}\left(\boldsymbol{\psi}\right) &= \mathcal{N}(\boldsymbol{\psi}; \mathbf{0}_{18}, \mathbf{I}), \\
\text{P}\left(\mathbf{z}_m | \boldsymbol{\mu}, \boldsymbol{\psi}\right) &= \mathcal{N}(\mathbf{z}_m; \boldsymbol{\mu}, \exp(\boldsymbol{\psi})\mathbf{I}) \\
\text{P}\left(\text{Rating}_{mn} | \mathbf{z}_m, \mathbf{x}_n\right) &= \text{Bernoulli}(\text{Rating}_{mn}; \sigma(\mathbf{z}_m^\mathsf{T}\mathbf{x}_n))
\end{aligned}
\tag{81}
$$

where $m \in \{1, \ldots, M\}$ and $n \in \{1, \ldots, N\}$. Note that we also have latent vectors for the global mean, $\boldsymbol{\mu}$, and log-variance, $\boldsymbol{\psi}$, of the weight vectors.

The factorised approximate posterior distribution $Q$ is initialised as:

$$
\begin{aligned}
\text{Q}\left(\boldsymbol{\mu}\right) &= \mathcal{N}(\boldsymbol{\mu}; \mathbf{0}_{18}, \mathbf{I}), \\
\text{Q}\left(\boldsymbol{\psi}\right) &= \mathcal{N}(\boldsymbol{\psi}; \mathbf{0}_{18}, \mathbf{I}), \\
\text{Q}\left(\mathbf{z}_m\right) &= \mathcal{N}(\mathbf{z}_m; \mathbf{0}_{18}, \mathbf{I})
\end{aligned}
\tag{82}
$$

To ensure high levels of uncertainty, we subsample the dataset to obtain a training set of $N = 5$ films and $M = 300$ users for our experiment. An equally sized but disjoint subset of users is held aside as a test set for calculation of the predictive log-likelihood.

---

3. Dataset and license: http://files.grouplens.org/datasets/movielens/ml-latest-small-README.html

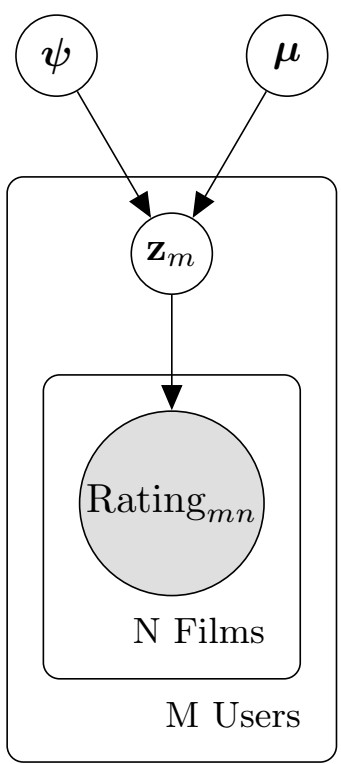

Figure 7: Graphical model for the MovieLens dataset

F.3.1. Reparameterized Movielens model

The reparameterized model is as follows,

$$
\begin{aligned}
\mathrm{P}\left(\boldsymbol{\mu}\right) &= \mathcal{N}(\boldsymbol{\mu}; \mathbf{0}_{18}, \mathbf{I}), \\
\mathrm{P}\left(\boldsymbol{\psi}\right) &= \mathcal{N}(\boldsymbol{\psi}; \mathbf{0}_{18}, \mathbf{I}), \\
\mathrm{P}\left(\mathbf{z}_m | \boldsymbol{\mu}, \boldsymbol{\psi}\right) &= \mathcal{N}(\mathbf{z}_m; \frac{\boldsymbol{\mu}}{100}, \frac{\exp(\boldsymbol{\psi})}{100}\mathbf{I}) \\
\mathrm{P}\left(\mathrm{Rating}_{mn} | \mathbf{z}_m, \mathbf{x}_n\right) &= \mathrm{Bernoulli}(\mathrm{Rating}_{mn}; \sigma(100 * \mathbf{z}_m^{\mathsf{T}} \mathbf{x}_n))
\end{aligned}
\tag{83}
$$

The corresponding factorised approximate posterior distribution $Q$ is then initialised as

$$
\begin{aligned}
\mathrm{Q}\left(\boldsymbol{\mu}\right) &= \mathcal{N}(\boldsymbol{\mu}; \mathbf{0}_{18}, \mathbf{I}), \\
\mathrm{Q}\left(\boldsymbol{\psi}\right) &= \mathcal{N}(\boldsymbol{\psi}; \mathbf{0}_{18}, \mathbf{I}), \\
\mathrm{Q}\left(\mathbf{z}_m\right) &= \mathcal{N}(\mathbf{z}_m; \mathbf{0}_{18}, \frac{1}{100}\mathbf{I})
\end{aligned}
\tag{84}
$$

## F.4. Occupancy dataset

The aim of occupancy modelling is to infer the presence of an animal at a given observation site from repeated samples, accounting for the possibilities of non-detection or false-detection (Doser et al., 2022). In this experiment we fit a modified multi-species occupancy model

to the North American Breeding Bird Survey data[4], which records over 700 species of bird across the contiguous United States and the 13 provinces and territories of Canada. The readings are collected along thousands of randomly selected road-side routes, taken at half mile intervals along the 24.5 mile routes for a total of 50 readings per route.

The survey is taken once per year, during peak breeding season, usually in June. The dataset we use covers the years 1966-2021, excluding 2020. We obtain R = 5 repeated samples from each route by taking the unit step function of the sum of every 10 readings (for a given route, year and species). From the main dataset, we obtain a training set by subsampling a set of $J = 12$ bird species, $M = 6$ years and $I = 200$ routes. A distinct test set for predictive log-likelihood calculation is obtained using the same values of $J$ and $M$, but a different set of $I = 100$ routes. Our model takes into consideration two sets of covariates: Weather$_{jmi}$ gives the temperature at site $i$ on year $m$ (replicated for each bird species); and Quality$_{jmi}$ indicates whether a particular series of readings at site $j$ on year $m$ followed all the recommended guidelines for recording birds.

We use a Bernoulli random variable to model the recording of a particular species $j$ of bird in a particular repeated measurement (given the route $i$ and year $m$) using logits which are calculated as the product of the quality covariate Quality$_{jmi}$, an inferred quality weight QualityWeight$_{jmi}$, and another inferred latent variable $z_{jmi}$ which indicates the true presence of bird species $j$ along route $i$ in year $m$, as well as including some probability of a false-positive bird sighting. This variable $z_{jmi}$ is also modelled by a Bernoulli distribution, with logits calculated as the product of the weather covariate Weather$_{jmi}$, an inferred weather weight WeatherWeight$_{jmi}$, and the latent variable BirdYearMean$_{jm}$ which represents the mean frequency of a specific species in a given year. Covariate weights, WeatherWeight$_{jmi}$ and QualityWeight$_{jmi}$, are modelled via separate Gaussian distributions, each of which have standard Gaussian hyperpriors for their mean and log-variance. The BirdYearMean$_{jm}$ variable also has an hierarchical prior: it comes from a Gaussian distribution with unit variance whose mean is the variable BirdMean$_j$, representing the mean frequency of species $j$ regardless of year or route, which in turn has its mean and log-variance modelled via standard Gaussian hyperpriors. The full model may be written as

---

4. Dataset: https://www.sciencebase.gov/catalog/item/625f151ed34e85fa62b7f926, licensing information is included with the dataset.

$$P\left(\mu_{\text{BirdMean}}\right) = \mathcal{N}(\mu_{\text{BirdMean}}; 0, 1),$$
$$P\left(\sigma_{\text{BirdMean}}\right) = \mathcal{N}(\sigma_{\text{BirdMean}}; 0, 1),$$
$$P\left(\mu_{\text{QualityWeight}}\right) = \mathcal{N}(\mu_{\text{QualityWeight}}; 0, 1),$$
$$P\left(\sigma_{\text{QualityWeight}}\right) = \mathcal{N}(\sigma_{\text{QualityWeight}}; 0, 1),$$
$$P\left(\mu_{\text{WeatherWeight}}\right) = \mathcal{N}(\mu_{\text{WeatherWeight}}; 0, 1),$$
$$P\left(\sigma_{\text{WeatherWeight}}\right) = \mathcal{N}(\sigma_{\text{WeatherWeight}}; 0, 1),$$
$$P\left(\text{QualityWeight}_j \middle| \mu_{\text{QualityWeight}}, \sigma_{\text{QualityWeight}}\right) = \mathcal{N}(\text{QualityWeight}_j; \mu_{\text{QualityWeight}},$$
$$\exp(\sigma_{\text{QualityWeight}}))$$
$$P\left(\text{WeatherWeight}_j \middle| \mu_{\text{WeatherWeight}}, \sigma_{\text{WeatherWeight}}\right) = \mathcal{N}(\text{WeatherWeight}_j; \mu_{\text{WeatherWeight}},$$
$$\exp(\sigma_{\text{WeatherWeight}}))$$
$$P\left(\text{BirdMean}_j | \mu_{\text{BirdMean}}, \sigma_{\text{BirdMean}}\right) = \mathcal{N}(\text{BirdMean}_j; \mu_{\text{BirdMean}},$$
$$\exp(\sigma_{\text{BirdMean}}))$$
$$P\left(\text{BirdYearMean}_{jm} | \text{BirdMean}_{jm}\right) = \mathcal{N}(\text{BirdYearMean}_{jm}; \text{BirdMean}_{jm}, 1)$$
$$\text{logits}_{jmi}^z = \text{BirdYearMean}_{jm} * \text{WeatherWeight}_j$$
$$* \text{Weather}_{jmi}$$
$$P\left(z_{jmi} \middle| \text{logits}_{jmi}^z\right) = \text{Bernoulli}(z_{jmi}; \text{logits} = \text{logits}_{jmi}^z)$$
$$\text{logits}_{jmir}^y = z_{jmi} * \text{QualityWeight}_j * \text{Quality}_{jmir}$$
$$+ (1 - z_{jmi}) * (-10)$$
$$P\left(y_{jmir} \middle| \text{logits}_{jmir}^y\right) = \text{Bernoulli}(y_{jmir}; \text{logits} = \text{logits}_{jmir}^y)$$

$$(85)$$

where $j \in \{1, \ldots, J\}, m \in \{1, \ldots, M\}, i \in \{1, \ldots, I\}$ and $r \in \{1, \ldots, R\}$. The corresponding graphical model is presented in Figure 8.

The factorised approximate posterior distribution $Q$ is initialised as:

$$Q\left(\mu_{\text{BirdMean}}\right) = \mathcal{N}(\mu_{\text{BirdMean}}; 0, 1),$$
$$Q\left(\sigma_{\text{BirdMean}}\right) = \mathcal{N}(\sigma_{\text{BirdMean}}; 0, 1),$$
$$Q\left(\mu_{\text{QualityWeight}}\right) = \mathcal{N}(\mu_{\text{QualityWeight}}; 0, 1),$$
$$Q\left(\sigma_{\text{QualityWeight}}\right) = \mathcal{N}(\sigma_{\text{QualityWeight}}; 0, 1),$$
$$Q\left(\mu_{\text{WeatherWeight}}\right) = \mathcal{N}(\mu_{\text{WeatherWeight}}; 0, 1),$$
$$Q\left(\sigma_{\text{WeatherWeight}}\right) = \mathcal{N}(\sigma_{\text{WeatherWeight}}; 0, 1),$$
$$Q\left(\text{QualityWeight}_j\right) = \mathcal{N}(\text{QualityWeight}_j; 0, 1)$$
$$Q\left(\text{WeatherWeight}_j\right) = \mathcal{N}(\text{WeatherWeight}_j; 0, 1)$$
$$Q\left(\text{BirdMean}_j\right) = \mathcal{N}(\text{BirdMean}_j; 0, 1)$$
$$Q\left(\text{BirdYearMean}_{jm}\right) = \mathcal{N}(\text{BirdYearMean}_{jm}; 0, 1)$$

$$(86)$$

F.4.1. Reparameterized occupancy model

The reparameterized occupancy model is as follows:

$$
\begin{aligned}
\mathrm{P}\left(\mu_{\text{BirdMean}}\right) &= \mathcal{N}(\mu_{\text{BirdMean}}; 0, 1), \\
\mathrm{P}\left(\sigma_{\text{BirdMean}}\right) &= \mathcal{N}(\sigma_{\text{BirdMean}}; 0, 1), \\
\mathrm{P}\left(\mu_{\text{QualityWeight}}\right) &= \mathcal{N}(\mu_{\text{QualityWeight}}; 0, 1), \\
\mathrm{P}\left(\sigma_{\text{QualityWeight}}\right) &= \mathcal{N}(\sigma_{\text{QualityWeight}}; 0, 1), \\
\mathrm{P}\left(\mu_{\text{WeatherWeight}}\right) &= \mathcal{N}(\mu_{\text{WeatherWeight}}; 0, 1), \\
\mathrm{P}\left(\sigma_{\text{WeatherWeight}}\right) &= \mathcal{N}(\sigma_{\text{WeatherWeight}}; 0, 1), \\
\mathrm{P}\left(\text{QualityWeight}_j \middle| \mu_{\text{QualityWeight}}, \sigma_{\text{QualityWeight}}\right) &= \mathcal{N}(\text{QualityWeight}_j; \mu_{\text{QualityWeight}}, \\
&\qquad \exp(\sigma_{\text{QualityWeight}})) \\
\mathrm{P}\left(\text{WeatherWeight}_j \middle| \mu_{\text{WeatherWeight}}, \sigma_{\text{WeatherWeight}}\right) &= \mathcal{N}(\text{WeatherWeight}_j; \mu_{\text{WeatherWeight}}, \\
&\qquad \exp(\sigma_{\text{WeatherWeight}})) \\
\mathrm{P}\left(\text{BirdMean}_j | \mu_{\text{BirdMean}}, \sigma_{\text{BirdMean}}\right) &= \mathcal{N}(\text{BirdMean}_j; \mu_{\text{BirdMean}}, \\
&\qquad \exp(\sigma_{\text{BirdMean}})) \\
\mathrm{P}\left(\text{BirdYearMean}_{jm} | \text{BirdMean}_{jm}\right) &= \mathcal{N}(\text{BirdYearMean}_{jm}; \\
&\qquad \frac{1}{1000}\text{BirdMean}_{jm}, \frac{1}{1000}) \\
\text{logits}^z_{jmi} &= 1000 * \text{BirdYearMean}_{jm} \\
&\qquad * \text{WeatherWeight}_j * \text{Weather}_{jmi} \\
\mathrm{P}\left(z_{jmi} \middle| \text{logits}^z_{\text{jmi}}\right) &= \text{Bernoulli}(z_{jmi}; \text{logits} = \text{logits}^z_{\text{jmi}}) \\
\text{logits}^y_{jmir} &= z_{jmi} * \text{QualityWeight}_j * \text{Quality}_{jmir} \\
&\qquad + (1 - z_{jmi}) * (-10) \\
\mathrm{P}\left(y_{jmir} \middle| \text{logits}^y_{\text{jmir}}\right) &= \text{Bernoulli}(y_{jmir}; \text{logits} = \text{logits}^y_{\text{jmir}})
\end{aligned}
\tag{87}
$$

The factorised approximate posterior distribution $Q$ is initialised as:

$$\begin{aligned}
\mathrm{Q}\left(\mu_{\mathrm{BirdMean}}\right) &= \mathcal{N}(\mu_{\mathrm{BirdMean}}; 0, 1), \\
\mathrm{Q}\left(\sigma_{\mathrm{BirdMean}}\right) &= \mathcal{N}(\sigma_{\mathrm{BirdMean}}; 0, 1), \\
\mathrm{Q}\left(\mu_{\mathrm{QualityWeight}}\right) &= \mathcal{N}(\mu_{\mathrm{QualityWeight}}; 0, 1), \\
\mathrm{Q}\left(\sigma_{\mathrm{QualityWeight}}\right) &= \mathcal{N}(\sigma_{\mathrm{QualityWeight}}; 0, 1), \\
\mathrm{Q}\left(\mu_{\mathrm{WeatherWeight}}\right) &= \mathcal{N}(\mu_{\mathrm{WeatherWeight}}; 0, 1), \\
\mathrm{Q}\left(\sigma_{\mathrm{WeatherWeight}}\right) &= \mathcal{N}(\sigma_{\mathrm{WeatherWeight}}; 0, 1), \\
\mathrm{Q}\left(\mathrm{QualityWeight}_j\right) &= \mathcal{N}(\mathrm{QualityWeight}_j; 0, 1) \\
\mathrm{Q}\left(\mathrm{WeatherWeight}_j\right) &= \mathcal{N}(\mathrm{WeatherWeight}_j; 0, 1) \\
\mathrm{Q}\left(\mathrm{BirdMean}_j\right) &= \mathcal{N}(\mathrm{BirdMean}_j; 0, 1) \\
\mathrm{Q}\left(\mathrm{BirdYearMean}_{jm}\right) &= \mathcal{N}(\mathrm{BirdYearMean}_{jm}; 0, \frac{1}{1000})
\end{aligned} \tag{88}$$

## F.5. Radon model

The radon dataset Price et al. (1996) consists of 12,777 home radon measurements in 9 states[5] Each reading is accompanied with information about which floor the reading was taken on, and county level soil uranium readings in parts per million. We take a subset of $S = 4$ states and $R = 300$ readings. We split this dataset in half, taking the results from 150 readings (per state) as our training set whilst the other 150 readings (per state) form our test set for the evaluation of predictive log-likelihood. The radon measurements are in log trillionth of a Curie per litre.

We model these readings as draws from a Gaussian distribution, with mean given by the sum of a per state mean $\mathrm{StateMean}_s$ and weighted basement and log uranium readings, and a per state variance $\mathrm{StateVariance}_s$. The weights and $\mathrm{StateVariance}_s$ are drawn from independent Normal distributions also in a per state fashion. The prior for $\mathrm{StateMean}_{sc}$ is given by a Gaussian distribution with a global mean GlobalMean and variance GlobalVariance, both of which are drawn from Normal distributions. A graphical model representation can be found in Figure 9. This can be written as

---

5. Dataset: http://www.stat.columbia.edu/~gelman/arm/examples/radon/ from Gelman and Hill (2006).

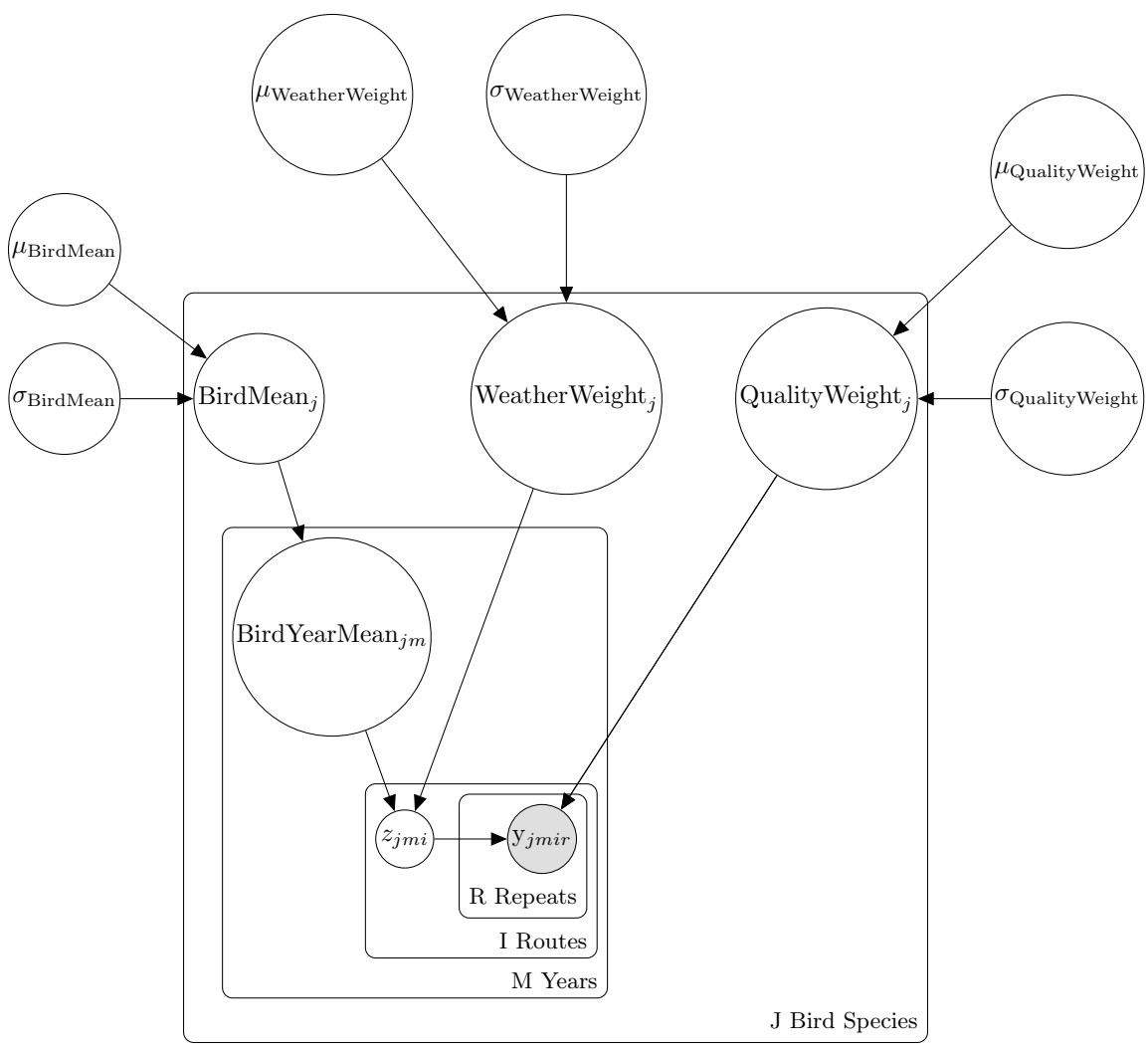

Figure 8: Graphical model for the Bird Occupancy dataset

$$P\left(\text{GlobalMean}\right) = \mathcal{N}(\text{GlobalMean}; 0, 1),$$

$$P\left(\text{GlobalVariance}\right) = \mathcal{N}(\text{GlobalVariance}; 0, 1),$$

$$P\left(\text{StateMean}_s | \text{GlobalMean}, \text{GlobalVariance}\right) = \mathcal{N}(\text{StateMean}_s; \text{GlobalMean},$$
$$\exp(\text{GlobalVariance}))$$

$$P\left(\text{StateVariance}_s\right) = \mathcal{N}(\text{StateVariance}_s; 0, 1)$$

$$P\left(\text{UraniumWeight}_s\right) = \mathcal{N}(\text{UraniumWeight}_{sc}; 0, 1)$$

$$P\left(\text{BasementWeight}_s\right) = \mathcal{N}(\text{BasementWeight}_s; 0, 1)$$

$$\text{RadonMean}_{sr} = \text{StateMean}_s$$
$$+ \text{UraniumWeight}_s * \text{Uranium}_s$$
$$+ \text{BasementWeight}_s * \text{Basement}_{sr}$$

$$P\left(\text{Radon}_{sr} | \text{RadonMean}_{sr}, \text{CountryVariance}_s\right) = \mathcal{N}(\text{Radon}_{sr}; \text{RadonMean}_{sr},$$
$$\exp(\text{StateVariance}_s))$$

$$(89)$$

where $s \in \{1, \ldots, S\}$ and $r \in \{1, \ldots, R\}$. We initialise our approximate posterior distribution $Q$ as follows:

$$
\begin{aligned}
Q\,(\text{GlobalMean}) &= \mathcal{N}(\text{GlobalMean}; 0, 1), \\
Q\,(\text{GlobalVariance}) &= \mathcal{N}(\text{GlobalVariance}; 0, 1), \\
Q\,(\text{StateMean}_s) &= \mathcal{N}(\text{StateMean}_s; 0, 1) \\
Q\,(\text{StateVariance}_s) &= \mathcal{N}(\text{StateVariance}_s; 0, 1) \\
Q\,(\text{UraniumWeight}_s) &= \mathcal{N}(\text{UraniumWeight}_s; 0, 1) \\
Q\,(\text{BasementWeight}_s) &= \mathcal{N}(\text{BasementWeight}_s; 0, 1)
\end{aligned}
\tag{90}
$$

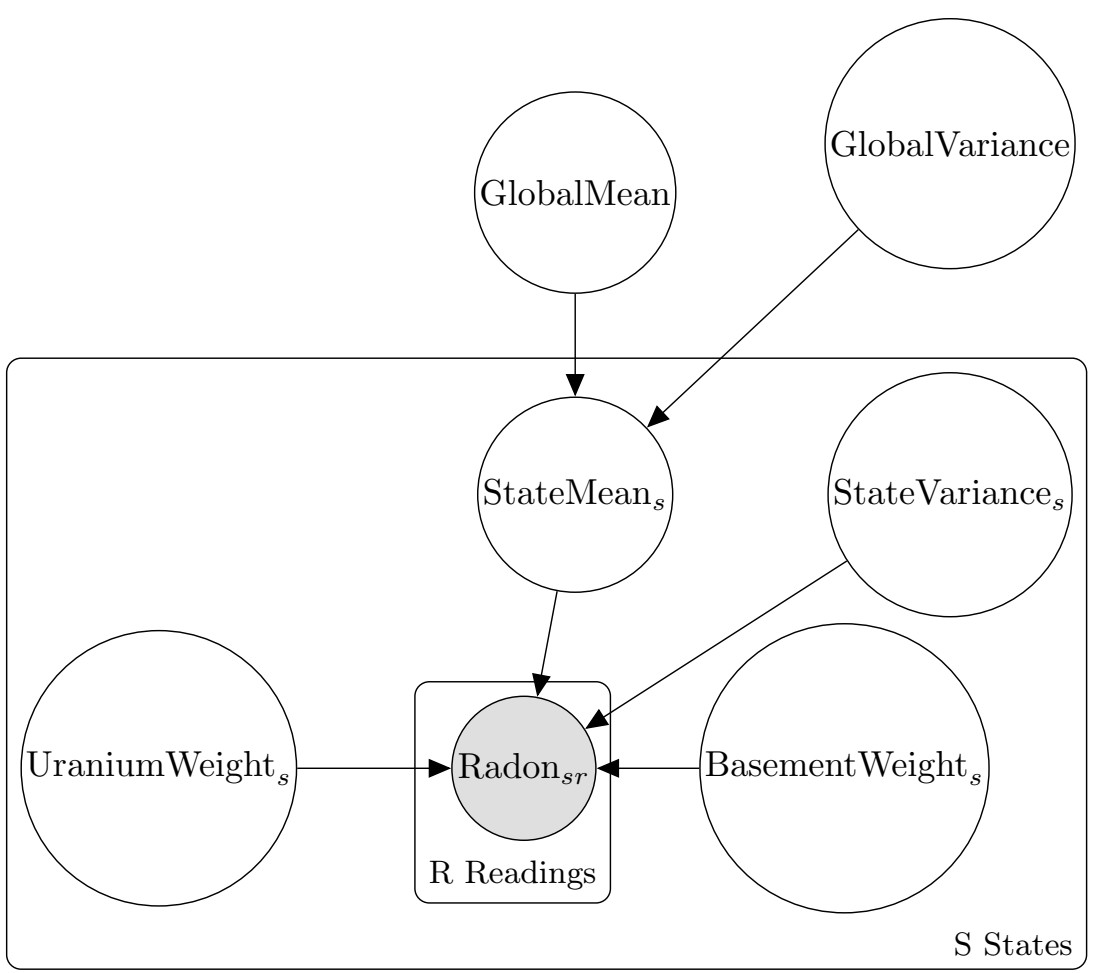

Figure 9: Graphical model for the Radon model

F.5.1. Reparameterized radon model

The reparameterized radon model is as so:

$$
\begin{aligned}
\mathrm{P}\,(\mathrm{GlobalMean}) &= \mathcal{N}(\mathrm{GlobalMean}; 0, 1), \\
\mathrm{P}\,(\mathrm{GlobalVariance}) &= \mathcal{N}(\mathrm{GlobalVariance}; 0, 1), \\
\mathrm{P}\,(\mathrm{StateMean}_s|\mathrm{GlobalMean}, \mathrm{GlobalVariance}) &= \mathcal{N}(\mathrm{StateMean}_s; \frac{\mathrm{GlobalMean}}{1000}, \\
&\qquad \frac{\exp(\mathrm{GlobalVariance})}{1000}) \\
\mathrm{P}\,(\mathrm{StateVariance}_s) &= \mathcal{N}(\mathrm{StateVariance}_s; 0, 1) \\
\mathrm{P}\,(\mathrm{UraniumWeight}_s) &= \mathcal{N}(\mathrm{UraniumWeight}_{sc}; 0, 1) \\
\mathrm{P}\,(\mathrm{BasementWeight}_s) &= \mathcal{N}(\mathrm{BasementWeight}_s; 0, 1) \\
\mathrm{RadonMean}_{sr} &= 1000 * \mathrm{StateMean}_s \\
&\quad + \mathrm{UraniumWeight}_s * \mathrm{Uranium}_s \\
&\quad + \mathrm{BasementWeight}_s * \mathrm{Basement}_{sr} \\
\mathrm{P}\,(\mathrm{Radon}_{sr}|\mathrm{RadonMean}_{sr}, \mathrm{CountryVariance}_s) &= \mathcal{N}(\mathrm{Radon}_{sr}; \mathrm{RadonMean}_{sr}, \\
&\qquad \exp(\mathrm{StateVariance}_s))
\end{aligned}
\tag{91}
$$

We initialise our approximate posterior distribution $Q$ as follows:

$$
\begin{aligned}
\mathrm{Q}\,(\mathrm{GlobalMean}) &= \mathcal{N}(\mathrm{GlobalMean}; 0, 1), \\
\mathrm{Q}\,(\mathrm{GlobalVariance}) &= \mathcal{N}(\mathrm{GlobalVariance}; 0, 1), \\
\mathrm{Q}\,(\mathrm{StateMean}_s) &= \mathcal{N}(\mathrm{StateMean}_s; 0, \frac{1}{1000}) \\
\mathrm{Q}\,(\mathrm{StateVariance}_s) &= \mathcal{N}(\mathrm{StateVariance}_s; 0, 1) \\
\mathrm{Q}\,(\mathrm{UraniumWeight}_s) &= \mathcal{N}(\mathrm{UraniumWeight}_s; 0, 1) \\
\mathrm{Q}\,(\mathrm{BasementWeight}_s) &= \mathcal{N}(\mathrm{BasementWeight}_s; 0, 1)
\end{aligned}
\tag{92}
$$

## F.6. Covid model

The Covid dataset we use consists of $D = 137$ daily Covid-19 infection measurements from $R = 92$ regions around the world Leech et al. (2022).[6] In addition the dataset contains region and day level information about average mobility, mask wearing, and other non-pharmaceutical interventions (NPIs). We model the observed Covid-19 infection numbers as a Negative Binomial random variable. The total count parameter of the Negative Binomial distribtion is sampled from a region level log-Normal distribution and the logits are given by a autoregressive Gaussian timeseries where, after the initial timestep which is sampled from

---

6. Dataset and license: https://github.com/g-leech/masks_v_mandates

a region level Gaussian distribution, each timestep is sampled from a Gaussian with a mean given by the sum of the previous timestep, a global mean term and a term depending on weighted average mobility, mask wearing and NPIs. The weights themselves are sampled from Gaussian distributions. The variance of this Gaussian distribution is sampled from a region level Gaussian, for which the mean and variance are sampled from global level Gaussian distributions.

We use the first $D = 109$ days of data as our training set and leave the remaining 28 days to form the test set on which we'll calculate predictive log-likelihood.

A graphical model representation of the model can be see in Figure 10. The model can be written as:

$$
\begin{aligned}
\mathrm{P}\left(\boldsymbol{\alpha_{\mathbf{NPIs}}}\right) &= \mathcal{N}(\alpha_{\mathrm{NPIs}}; \mathbf{0}_9, \boldsymbol{I}_9), \\
\mathrm{P}\left(\alpha_{\mathrm{Wearing}}\right) &= \mathcal{N}(\alpha_{\mathrm{Wearing}}; 0, 0.4), \\
\mathrm{P}\left(\alpha_{\mathrm{Mobility}}\right) &= \mathcal{N}(\alpha_{\mathrm{Mobility}}; 1.704, 0.44)) \\
\mathrm{P}\left(\mathrm{GlobalMean}\right) &= \mathcal{N}(\mathrm{GlobalMean}; 1.07, 0.6) \\
\mathrm{P}\left(\mathrm{InitialSizeMean}\right) &= \mathcal{N}(\mathrm{InitialSizeMean}; \log(1000), 0.5) \\
\mathrm{P}\left(\mathrm{InfectedNoiseMean}\right) &= \mathcal{N}(\mathrm{InfectedNoiseMean}; \log(0.01), 0.25) \\
\mathrm{P}\left(\mathrm{MeanInfected}_{r,1} | \mathrm{InitialSizeMean}\right) &= \mathcal{N}(\mathrm{MeanInfected}_{r,1}; \\
&\quad \mathrm{InitialSizeMean}, 0.5) \\
\mathrm{P}\left(\mathrm{InfectedNoise}_r | \mathrm{InfectedNoiseMean}\right) &= \mathcal{N}(\mathrm{InfectedNoise}_r; \\
&\quad \mathrm{InfectedNoiseMean}, 0.25) \\
\mathrm{P}\left(\psi_r\right) &= \mathcal{N}(\psi_r; 0, 1) \\
\mathrm{NPI}_{r,d} &= \mathrm{GlobalMean} + \boldsymbol{\alpha_{\mathbf{NPIs}}}\mathrm{NPIs}_{r,d} \\
&\quad + \alpha_{\mathrm{Wearing}} * \mathrm{Wearing}_{r,d} \\
&\quad + \alpha_{\mathrm{Mobility}} * \mathrm{Mobility}_{r,d} \\
\mathrm{NewMean}_{r,d} &= \mathrm{NPI}_{r,d} + \mathrm{MeanInfected}_{r,d-1} \\
\mathrm{P}\left(\mathrm{MeanInfected}_{r,d} | \mathrm{NewMean}_{r,d}, \mathrm{InfectedNoise}_r\right) &= \mathcal{N}(\mathrm{MeanInfected}_{r,d}; \mathrm{NewMean}_{r,d}, \\
&\quad \exp(\mathrm{InfectedNoise}_r)) \\
\mathrm{P}\left(\mathrm{Infected}_{r,d} | \psi, \mathrm{MeanInfected}_{r,d}\right) &= \mathrm{NegativeBinomial}(\mathrm{Infected}_{r,d}; \exp(\psi), \\
&\quad \exp(\psi - \mathrm{MeanInfected}_{r,d}) + 1 \\
&\quad + 10^{-7})
\end{aligned}
\tag{93}
$$

where $r \in \{1, \ldots, R\}$ and $d \in \{2, \ldots, D\}$ (note that the initial time step, $d = 1$, is dealt with separately from subsequent time steps). We initialise our approximate posterior distribution $Q$ as follows:

$$Q\left(\boldsymbol{\alpha}_{\mathbf{NPIs}}\right) = \mathcal{N}(\alpha_{\text{NPIs}}; \mathbf{0}_9, \boldsymbol{I}_9),$$
$$Q\left(\alpha_{\text{Wearing}}\right) = \mathcal{N}(\alpha_{\text{Wearing}}; 0, 1),$$
$$Q\left(\alpha_{\text{Mobility}}\right) = \mathcal{N}(\alpha_{\text{Mobility}}; 0, 1))$$
$$Q\left(\text{GlobalMean}\right) = \mathcal{N}(\text{GlobalMean}; 0, 1)$$
$$Q\left(\text{InitialSizeMean}\right) = \mathcal{N}(\text{InitialSizeMean}; 0, 1)$$
$$Q\left(\text{InfectedNoiseMean}\right) = \mathcal{N}(\text{InfectedNoiseMean}; 0, 1) \tag{94}$$
$$Q\left(\text{MeanInfected}_{r,1}\right) = \mathcal{N}(\text{MeanInfected}_{r,1}; 0, 1)$$
$$Q\left(\text{InfectedNoise}_r\right) = \mathcal{N}(\text{InfectedNoise}_r; 0, 1)$$
$$Q\left(\psi_r\right) = \mathcal{N}(\psi_r; 0, 1)$$
$$Q\left(\text{MeanInfected}_{r,d}\right) = \mathcal{N}(\text{MeanInfected}_{r,d}; 0, 1)$$

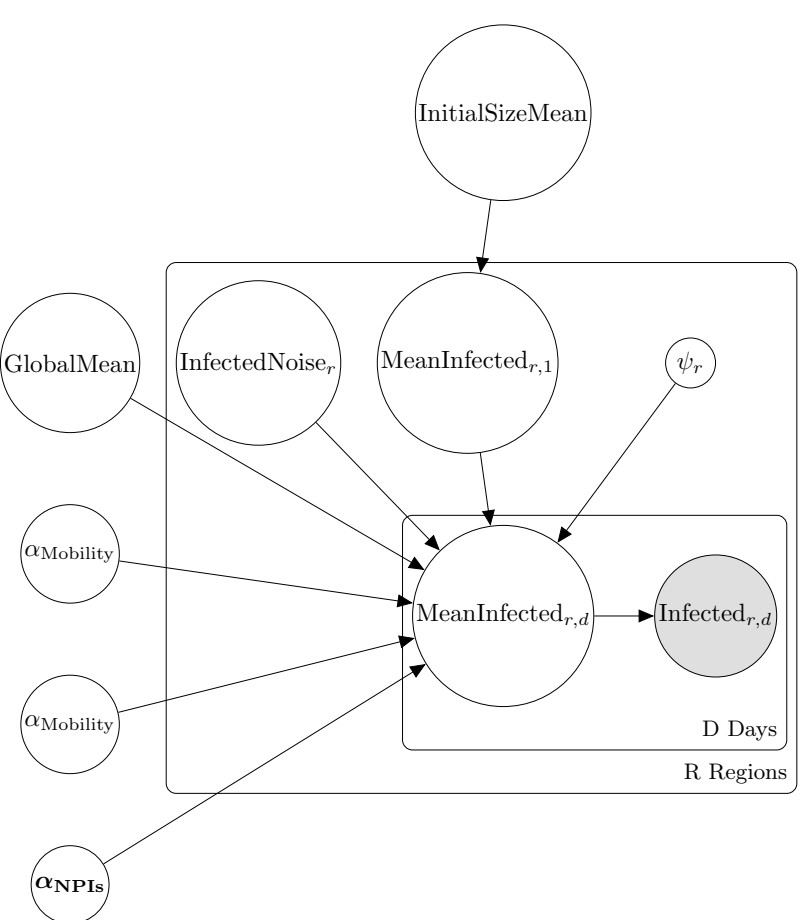

Figure 10: Graphical model for the Covid model

F.6.1. Reparameterized covid model

The reparameterized covid model uses the following generative distribution

$$
\begin{aligned}
\mathrm{P}\left(\boldsymbol{\alpha_{\mathbf{NPIs}}}\right) &= \mathcal{N}(\alpha_{\mathrm{NPIs}}; \mathbf{0}_9, \boldsymbol{I}_9), \\
\mathrm{P}\left(\alpha_{\mathrm{Wearing}}\right) &= \mathcal{N}(\alpha_{\mathrm{Wearing}}; 0, \frac{0.4}{10000}), \\
\mathrm{P}\left(\alpha_{\mathrm{Mobility}}\right) &= \mathcal{N}(\alpha_{\mathrm{Mobility}}; 1.704, 0.44)) \\
\mathrm{P}\left(\mathrm{GlobalMean}\right) &= \mathcal{N}(\mathrm{GlobalMean}; 1.07, 0.6) \\
\mathrm{P}\left(\mathrm{InitialSizeMean}\right) &= \mathcal{N}(\mathrm{InitialSizeMean}; \log(1000), 0.5) \\
\mathrm{P}\left(\mathrm{InfectedNoiseMean}\right) &= \mathcal{N}(\mathrm{InfectedNoiseMean}; \log(0.01), 0.25) \\
\mathrm{P}\left(\mathrm{MeanInfected}_{r,1}|\mathrm{InitialSizeMean}\right) &= \mathcal{N}(\mathrm{MeanInfected}_{r,1}; \\
&\qquad \mathrm{InitialSizeMean}, 0.5) \\
\mathrm{P}\left(\mathrm{InfectedNoise}_r|\mathrm{InfectedNoiseMean}\right) &= \mathcal{N}(\mathrm{InfectedNoise}_r; \\
&\qquad \mathrm{InfectedNoiseMean}, 0.25) \\
\mathrm{P}\left(\psi_r\right) &= \mathcal{N}(\psi_r; 0, 1) \\
\mathrm{NPI}_{r,d} &= \mathrm{GlobalMean} + \boldsymbol{\alpha_{\mathbf{NPIs}}}\mathrm{NPIs}_{r,d} \\
&\qquad + 10000 * \alpha_{\mathrm{Wearing}} * \mathrm{Wearing}_{r,d} \\
&\qquad + \alpha_{\mathrm{Mobility}} * \mathrm{Mobility}_{r,d} \\
\mathrm{NewMean}_{r,d} &= \mathrm{NPI}_{r,d} + \mathrm{MeanInfected}_{r,d-1} \\
\mathrm{P}\left(\mathrm{MeanInfected}_{r,d}|\mathrm{NewMean}_{r,d}, \mathrm{InfectedNoise}_r\right) &= \mathcal{N}(\mathrm{MeanInfected}_{r,d}; \mathrm{NewMean}_{r,d}, \\
&\qquad \exp(\mathrm{InfectedNoise}_r)) \\
\mathrm{P}\left(\mathrm{Infected}_{r,d}|\psi, \mathrm{MeanInfected}_{r,d}\right) &= \mathrm{NegativeBinomial}(\mathrm{Infected}_{r,d}; \exp(\psi), \\
&\qquad \exp(\psi - \mathrm{MeanInfected}_{r,d}) + 1 \\
&\qquad + 10^{-7})
\end{aligned}
\tag{95}
$$

We initialise the reparameterized approximate posterior distribution $Q$ as follows:

$$\mathrm{Q}\left(\boldsymbol{\alpha_{\mathbf{NPIs}}}\right) = \mathcal{N}(\alpha_{\mathrm{NPIs}}; \mathbf{0}_9, \boldsymbol{I}_9),$$

$$\mathrm{Q}\left(\alpha_{\mathrm{Wearing}}\right) = \mathcal{N}(\alpha_{\mathrm{Wearing}}; 0, \frac{1}{10000}),$$

$$\mathrm{Q}\left(\alpha_{\mathrm{Mobility}}\right) = \mathcal{N}(\alpha_{\mathrm{Mobility}}; 0, 1))$$

$$\mathrm{Q}\left(\mathrm{GlobalMean}\right) = \mathcal{N}(\mathrm{GlobalMean}; 0, 1)$$

$$\mathrm{Q}\left(\mathrm{InitialSizeMean}\right) = \mathcal{N}(\mathrm{InitialSizeMean}; 0, 1)$$

$$\mathrm{Q}\left(\mathrm{InfectedNoiseMean}\right) = \mathcal{N}(\mathrm{InfectedNoiseMean}; 0, 1) \tag{96}$$

$$\mathrm{Q}\left(\mathrm{MeanInfected}_{r,1}\right) = \mathcal{N}(\mathrm{MeanInfected}_{r,1}; 0, 1)$$

$$\mathrm{Q}\left(\mathrm{InfectedNoise}_r\right) = \mathcal{N}(\mathrm{InfectedNoise}_r; 0, 1)$$

$$\mathrm{Q}\left(\psi_r\right) = \mathcal{N}(\psi_r; 0, 1)$$

$$\mathrm{Q}\left(\mathrm{MeanInfected}_{r,d}\right) = \mathcal{N}(\mathrm{MeanInfected}_{r,d}; 0, 1)$$

## Appendix G. Train-Test Splits for Experimental Evaluation

For our experimental evaluation, we carefully constructed train-test splits for each dataset to ensure robust assessment of model performance. Below we detail the specific splits used for each model:

### G.1. Bus Breakdown Dataset

The NYC Bus Breakdown dataset was subsampled to include $Y = 2$ years and $B = 3$ boroughs. For each borough-year combination, we constructed a training set of $I = 150$ delayed buses and an equally-sized test set of an additional 150 delayed buses for calculating predictive log-likelihood. Each delay was uniquely identified by the year, borough, and index.

### G.2. MovieLens Dataset

From the MovieLens100K dataset, we constructed a training set consisting of $N = 5$ films and $M = 300$ users. An equally sized but completely disjoint subset of users was held out as a test set for evaluating predictive log-likelihood. All ratings were binarized, with user ratings of $\{0, 1, 2, 3\}$ mapped to 0 (dislikes) and ratings of $\{4, 5\}$ mapped to 1 (likes).

### G.3. Bird Occupancy Dataset

For the North American Breeding Bird Survey data, we created a training set containing $J = 12$ bird species, $M = 6$ years, and $I = 200$ routes. The test set for predictive log-likelihood calculation used the same species and years but a different set of 100 routes. We obtained $R = 5$ repeated samples from each route by taking the unit step function of the sum of every 10 readings along each route.

### G.4. Radon Dataset

We used a subset of the radon measurement dataset consisting of $S = 4$ states and $R = 300$ readings per state. This was split evenly, with 150 readings per state forming the training set and the other 150 readings per state constituting the test set for evaluating predictive log-likelihood.

### G.5. Covid Dataset

The Covid dataset contained $D = 137$ daily Covid-19 infection measurements from $R = 92$ regions worldwide. The first 109 days of data were used as the training set, while the remaining 28 days formed the test set for calculating predictive log-likelihood. This temporal split allows us to evaluate the model's predictive performance on future infection rates.

