# OpenReview forum: "Massively Parallel Expectation Maximization For Approximate Posteriors"
_approximateinference.org/AABI/2025/Proceedings_Track — AABI 2025 Proceedings Track_

### Official Review · Reviewer_JFW1 · 2025-02-24
**An incremental contribution to a recent series of massively parallel computing algorithms**

**Rating:** 6
**Confidence:** 3

**Review:**

This paper introduces a Massively Parallel Expectation Maximization (MPEM) method for approximating posterior distributions. The approach is inspired by a recent series of massively parallel computing algorithms, including massively parallel importance weighting (Bowyer et al., 2024), massively parallel variational inference (VI, Aitchison, 2019), and massively parallel reweighted wake-sleep (RWS, Heap et al., 2023). Experimental results demonstrate that the proposed method outperforms state-of-the-art, massively parallel variants of RWS and VI, while maintaining robustness to reparameterizations of the model that typically slow down gradient-based methods.

---

### Official Review · Reviewer_2S8q · 2025-02-25
**Moment-matching-based adaptive importance sampling**

**Rating:** 7
**Confidence:** 3

**Review:**

## Overview

Let
$$Q(z;\phi)\propto \exp\left(\sum_{j=1}^M T_j(z)\phi_j\right)$$
denote an exponential family with $M$ sufficient statistics.  We seek to identify $\phi^*$ so that $Q(\cdot;\phi^*)$ approximates a posterior of interest, $P(z|x)$.  The article under review proposes the following algorithm.

Initialize $\mu \in \mathbb{R}^M$.  Until bored, repeat:
1. **Calculate natural parameters,** identifying $\phi$ so that $E_{Z \sim Q(\cdot;\phi)}[T_j(Z)]=\mu_j$ for all $j$.
2. **Re-estimate posterior moments,** $\mu_{j}' \approx E_{(Z,X)\sim P}[T_j(Z)|X=x]$ for all $j$.  Expectations are approximated using MPIW with $Q(\cdot;\phi)$ as a proposal distribution.
3. **Update moment estimates with an exponential moving average,** $\mu \gets \lambda \mu' + (1-\lambda) \mu$

The resulting algorithm appears to perform reasonably well on a nice suite of benchmarks.

## Contributions

The idea seems good.  In fact I am surprised the general approach hasn't been done before, but I wasn't able to find cite, so maybe it really is new.  It's quite similar to cross-entropy-based adaptive importance sampling methods with exponential family proposals, but keeping a moving average of the moments seems to be a genuinely new contribution to the adaptive importance sampling zeitgeist.

In my view, the overall loop of

* update posterior moment estimates, using proposal distribution
* update proposal distribution by moment-matching (with exponential moving average)

is the strongest contribution.  It seems like quite a general-purpose approach.

## Concerns

My top concern is that some of the the strengths and limitations of the new approach seem a bit unclear from the manuscript.

**Strengths that aren't made clear**
* The new overall approach could be suitable even without using MPIW for getting moments.   Likelihood-tempered SMC, MCMC (with Q as proposal), or even plain vanilla importance sampling might work great.  The new overall approach actually isn't complicated at all, and could be implemented quite easily, whereas with the MPIW it becomes something that (at least with packages in their current state) requires Serious Heavy Lifting to implement.

**Weaknesses that aren't made clear**
* The $O(K^{|\mathrm{qa}(i)|})$ compute problem is a bit buried, and not written in a way that is particularly clear for the innocent reader ☺.  The basic story isn't hard to grasp: MPIW only works if your distributions factor according to a Bayes net, and that Bayes net can't includes a lot of parents.  If there's a lot of parents, MPIW is off the table.  I think this is true of both the prior and the proposal distributions: neither of them can need Bayes nets with lots of parents.  So, even with mean field proposal distributions, I think (though I'm not sure!) that this won't work if the prior has very messy dependencies.  By contrast, as it is written, it isn't even immediately obvious to the reader that you need a Bayes net at all.  It would be nice if the situation was more up-front.
* Related to above, right now the article has a remark like "this works especially well with exponential families!"  But I think in practice it would be simpler to just write "This article assumes exponential families."  All experiments and so-on are done that way as far as I can tell.  And then you can hypothesize that in future it might be extended beyond.  As is, it runs the risk of appearing to be a more general contribution than has actually been made.
* As you no doubt know, the moment estimates are *not* unbiased (except in the limit as $K\rightarrow \infty$).  They're self-normalized importance samples after all.  So then Theorem 1 doesn't *exactly* apply.  The theorem is still worth including, of course, but I think the reader could use a reminder that in most cases importance sampling can only get you unbiased estimates for the partition function, nothing else.
* Approaches with RWS, ELBO, etc have a clear path to estimating the generative model as well as estimating an approximate posterior.  Here there might be such a path but, at least for now, it is obscure.  I think reader could use to have that clarified for them somewhere in the article: we're really just doing Bayesian inference and not learning a generative model (even though some of the competitors could be used to learn a generative model).
* In practice, there's no real loss function that you know "should" go down at each step.  It's possible this algorithm will just chase its tail.  This may make it harder to debug, at the very least.  This is alluded to, but not made crystal clear.

A few additional concerns.
* I wasn't able to figure out the exact meaning of the predictive log-likelihood, as it is actually defined for each of the datasets.  For Movielens it was clearly stated in the appendix, but for example, for Bus breakdown, how was the test/train split organized?  It might be in there and I just missed it.  But if you could highlight the test/train splits more clearly it would be helpful.
* It wouldn't be a review unless I suggested additional simulations ☺.  Comparisons with a wider variety of techniques would be welcome.  For example, old-fashioned ELBO minimization and forward-amortized variational inference would be direct competitors.  To the extent these posteriors are used for estimating expectations, vanilla importance sampling would also be a natural competitor (which presumably you would trounce, but it would be nice to see that this is actually true as a function of compute time... you could handle a lot of vanilla importance sampling particles in 50 seconds on a GPU!).  The MCMC currently used to compute approximate "truth" would also be a competitor: how long did the "long runs" take to converge within the same margin of error as QEM?

## Quibbles

* "qa" is used well before it is defined.
* The remark "Note we cannot run VI on the occupancy model as it has discrete latent variables" makes it sound like variational inference as a whole somehow is incapable of handling discrete latent variables ☺.  Perhaps "Note we did not run MP VI on the occupancy model as it has discrete latent variables" would be less confusing.

## Last minute edit

Sorry for this late addendum to the review, it just occurred to me much later.

For, e.g., a normal distribution proposal, are you tracking mean and variance as moments or are you tracking E[X] and E[X^2]?  And, relatedly, if K is too small do you ever find that the variance moments get estimated as Very Very small due to a small number of particles getting all the weight?  It seems like this could yield major failure of the algorithm: proposal distribution collapses to a point.  Did you find this could happen?

---

### Official Review · Reviewer_wNou · 2025-02-28
**Review of "Massively Parallel Expectation Maximization For Approximate Posteriors"**

**Rating:** 7
**Confidence:** 3

**Review:**

In this paper, the authors tackle the task of Bayesian inference for complex hierarchical models by proposing a novel algorithm, named "Q Expectation-Maximization" (QEM), that aims to learn an approximate posterior chosen among an exponential family. In contrast to standard Variational Inference (VI) or Reweighted Wake-Sleep (RWS) methods, the proposed approach does no rely on a gradient descent scheme (that may be costly). It rather iteratively fits the approximate posterior based on accurate moment estimations of the true posterior, obtained via massively parallel importance weighting technique. In addition to being well justified in theory, this method outperforms numerically massively parallel VI and RWS methods on a large benchmark through various metrics (ELBOs, predictive log-likelihood...) while taking less computational budget.

The paper is very well written, and is enjoyable to read. In particular, the motivation of the method's design is well exposed and justified, and the idea of combining the EM scheme with the MPIW setting for approximating posteriors is highly relevant. What is overall remarkable is the perceived simplicity of the framework, which turns into a robust and accurate algorithm in practice. Interestingly, QEM does not seem to require specific hyperparameter tuning, which can be a practical burden in realistic sampling tasks. This is a great paper.

---

### Official Review · Reviewer_hoZ9 · 2025-02-28
**Interesting and possibly efficient but challenging to follow due to poor presentation**

**Rating:** 6
**Confidence:** 3

**Review:**

The manuscript is challenging to read, and it seems necessary to consult
[1, 2]
and possibly [@aitchisonTensorMonteCarlo2019] to fully understand the
paper. The presentation suffers from a lack of structure, leading to
repetitions and an unintuitive flow. For instance, the definition of
$qa(i)$ in equation 6b is not immediately clear, and the concept of
conditional independence a posteriori is not introduced beforehand. A
simple explanation that the authors target posteriors with a specific
dependence structure, maybe with a reference to one of the Directed
Acycle Graphs presented in the appendix, would help.

The idea itself is intriguing and appears effective across various
examples previously analysed in other papers. It could be interesting to
discuss whether, besides convergence to the correct moments, the method
also converges to the best posterior approximation in a Kullback-Leibler
sense or another metric. This might hold only for exponential family
models, where the method seems most practical, but it would be
interesting enough already.

The reparametrisation invariance initially appears restrictive, covering
only a very specific reparametrisation (scaling). However, this aspect
is interesting, and Figure 5 demonstrates its added value.

- The acronym QEM presumably stands for Quasi Expectation
    Maximization, but as the name of the method, it should definitely be
    explicitly defined in the text.

- The article Chatterjee and Diaconis (2017) is cited multiple times with the same
    intent; consider removing some of these recurring references.

- That is because EP uses an elaborate update procedure involving
    taking a replacing a factor $\rightarrow$ That is because EP uses an
    elaborate update procedure involving replacing a factor

- In section 3.2, it would be clearer to first explain that the
    distribution Q is an importance sampling proposal and an
    approximate posterior is chosen to create a good proposal. This
    context makes the $K$ samples more understandable, as they are
    sampled from the proposal, and equations 6a to 6c become clearer. At
    the moment, we do not know what they are sampled from.

- Correct the sentence: \"We can compute the limit inside the
    expoential using the L'Hoptial's rule\" to \"We can compute the
    limit inside the exponential using L'Hopital's rule.\"

- Equations 12 and 16 are incompatible; use superscripts consistently
    and consider removing one of the equations.

- Appendix A introduces \"global\" without discussing a local
    alternative. Provide more details to avoid requiring the reader to
    refer to Geffner and Domke.

- Appendix B lacks clarity and should be revised for better
    understanding.

## References

1. Thomas Heap, Gavin Leech, and Laurence Aitchison. *Massively parallel reweighted wake-sleep*. In *Proceedings of the Thirty-Ninth Conference on Uncertainty in Artificial Intelligence*, pages 870–878. PMLR, July 2023. [@heapMassivelyParallelReweighted2023].

2. Sam Bowyer, Thomas Heap, and Laurence Aitchison. *Using Autodiff to Estimate Posterior Moments, Marginals and Samples*. In *Proceedings of the Fortieth Conference on Uncertainty in Artificial Intelligence*, pages 394–417. PMLR, September 2024. [@bowyerUsingAutodiffEstimate2024].

3. Laurence Aitchison. *Tensor Monte Carlo: Particle Methods for the GPU era*. In *Advances in Neural Information Processing Systems*, volume 32. Curran Associates, Inc., 2019. [@aitchisonTensorMonteCarlo2019].

---

### Official Review · Reviewer_Z2Fo · 2025-02-28
**The paper proposes a variant of the expectation maximization method named QEM, which purportedly outperforms Variational Inference (VI) and Reweighted Wake-Sleep (RWS) as demonstrated in their experiments. However, the novelty of the method is not entirely convincing.**

**Rating:** 4
**Confidence:** 5

**Review:**

The QEM algorithm approximates the moments: expectation $E_{P(z|x)}[m(z)]$ where $m(z)$ corresponds to a moment function, e.g., first or second moments corresponding to $m(z) = z$ or $m(z) = z^2$. The key idea is to use massively parallel approximation importance weighting (MPIW) of the posterior distribution to improve the posterior approximation of the desired expectation (moments) beyond  simpler approximation methods like importance sampling.

This algorithm seems to fall under the umbrella of Monte Carlo Expectation Maximization algorithm, first proposed by Wei and Tanner (1990). In the MCEM, the E-step is carried out by sampling from the posterior $Z | X, \theta^{(t-1)}$, using a Monte Carlo algorithm because the exact E-step is difficult to compute. The Monte Carlo algorithm that can be utilized within MCEM includes importance sampling, Markov chain Monte Carlo, or more complex methods such as sequential Monte carlo methods and other variants such as stochastic EM. Therefore, the idea of QEM to use MPIW places it under the MCEM framework as MPIW is essentially an importance sampling algorithm.

There is a critical inaccuracy in the following statement:

"Importantly, QEM differs from standard EM because in standard EM, there is no approximate posterior. Instead, standard EM learns parameters of the model $\theta$ which may appear in $P_{\theta}(z')$ or in the likelihood, $P_{\theta}(x|z')$."

This statement misrepresents the EM algorithm, where the E-step of the EM algorithm indeed involves computing the expectation with respect to the posterior $Z | X, \theta^{(t-1)}$, where $\theta^{(t-1)}$ is the model parameters from previous iteration. In fact, MC-EM was developed to approximate this very posterior because the exact E-step is intractable.

In relation to the above point, algorithm 1 is missing a clear M-step. The M-step of the EM algorithm optimizes the resulting expression from the E-step. So I wonder how the model parameters $\theta$ are updated. In appendix G, it looked like the parameters were pre-set and not updated, which diverges from the typical iterative nature of EM algorithms.

The motivation cited from Chatterjee and Diaconis (2018) concerns the exponential number of importance samples required as the number of latent variables increases. But I am not convinced that the approach proposed in thie paper addresses this concern. With sufficiently large $K$, computing the summation over the indices in Eq~(6) will be time consuming at $O(K^n)$. In simple cases, where the latent variables not interacting in some complex manner, e.g., conditional independence, then the number of samples $K$ may be kept minimial. But outside of these cases, I am not quite sure how this approach solves the need for an exponential number importance samples as the latent variable size increases.

---

### Official Review · Reviewer_Jnjm · 2025-03-01
**excellent work**

**Rating:** 9
**Confidence:** 4

**Review:**

This paper is fully complete in both theory and experiments, which is an excellent article, and the proposed method has wide value in the application of Graphical Models.

Look forward to the open source code, and related methods of comparison.

---

### Meta-Review · Area_Chair_G2dq · 2025-03-17

**Recommendation:** Accept
**Confidence:** 4

**Metareview:**

The authors propose QEM, a method that leverages massively parallel importance weighting to efficiently estimate posterior moments and fit an approximate posterior via expectation maximization, achieving faster performance than state-of-the-art alternatives while being invariant to model reparameterizations. The reviewers are in overall agreement on acceptance, and I recommend that the authors take into account the reviewer points and implement these in the camera-ready.

---

### Decision · Program_Chairs · 2025-03-18

Accept